



# Multi-fidelity actuator line modelling of tandem floating offshore wind turbines

Agnese Firpo [1], Andrea Giuseppe Sanvito [1], Giacomo Persico [1], and Vincenzo Dossena [1]

[1]Department of Energy, Politecnico di Milano, Via Lambruschini 4, Milano, 20156, Italy

**Correspondence:** Giacomo Persico (giacomo.persico@polimi.it)

**Abstract.** The motion of floating offshore wind turbine platforms strongly affects wake development, influencing energy production, farm layout, and turbine loads. Unlike fixed-bottom turbines, floating turbine wake interactions are less understood and require high-fidelity modeling. This study employs an Actuator Line Model approach to investigate two turbines, with the upstream platform undergoing surge and pitch motions. Multi-fidelity simulations (URANS, LES with laminar and turbulent inflow) are performed to separate the effects of platform motion and inflow turbulence on wake dynamics and to assess URANS capability in capturing floating turbine wakes. Simulations of the single floating turbine are validated against experimental load and wake data. Wake validation shows that LES with turbulent inflow best captures turbulence intensity distribution, while URANS reproduces mean velocity profiles accurately especially in the near-wake. Platform motion enhances wake recovery under laminar inflow, whereas under turbulent inflow the wake recovers faster and the effect of motion is reduced; URANS shows slower far-wake recovery compared to turbulent LES. Analysis of platform-motion-induced wake oscillations indicates that turbulent LES accurately reproduces amplitudes, while URANS underestimates them and laminar inflow LES is inadequate. Furthermore, no significant differences is found between surge and pitch cases. Wake meandering is found to be primarily driven by turbulent inflow rather than platform motion: turbulent LES captures wake displacement at a characteristic frequency, whereas URANS fails to reproduce it. The impact of the wake on a downstream turbine 5D away is finally assessed and reveals that URANS underestimates both mean values and amplitudes of the downstream turbine loads. Blade distributed load analysis shows that URANS captures platform-motion-induced variability upstream but misses turbulence-driven effects downstream. In conclusion, this study provides a detailed characterization of floating turbine wake dynamics, highlights the different accuracy of LES and URANS, and demonstrates that LES yields more reliable predictions of downstream turbine loads, essential for their structural assessment within wind farms.

## 1 Introduction

The rapid expansion of offshore wind energy has driven the industry to explore locations farther from shore, aiming to harness the great potential of stronger and more uniform winds. Therefore, deeper waters require the use of the most suitable floating offshore wind turbines (FOWTs). Unlike fixed-bottom turbines, however, FOWTs are free to move according to six degrees of freedom, namely influenced by the combined effects of wind, waves, and currents. The motion of floating platforms influences the development of wake structures, suggesting that wake interactions within floating wind farms may differ significantly



from those in conventional fixed-bottom installations. Consequently, investigating these interactions is crucial, as they have a direct impact on overall energy production, optimal farm layout, and turbine longevity due to structural fatigue. While wake interactions among traditional fixed-bottom turbines have been widely studied, those involving floating wind turbines remain less explored and not yet fully understood, partly due to their greater complexity.

The dynamic behaviour of floating wind turbines is particularly complex due to phenomena such as unsteady aerodynamics, which can influence turbine loading at specific reduced frequencies of platform motion (Schulz et al. (2024), Sanvito et al. (2024b)). An additional layer of complexity in wake dynamics is introduced by modern active dynamic control strategies, such as Dynamic Induction Control (DIC) and Dynamic Individual Pitch Control (DIPC). In the case of floating turbines, ongoing research is examining how these time-varying control strategies affect platform motion and, consequently, wake evolution

(Berg et al. (2022), van den Berg et al. (2023)).

    In parallel, other aspects of wake dynamics remain open research questions. One key example is the potential influence of platform motion on wake meandering mechanisms. Li et al. (2022) has shown that side-to-side platform motion can induce wake meandering under low inflow turbulence conditions. Messmer et al. (2024a) has experimentally demonstrated that a platform movement aligned with the wind direction (surge motion) can affect wake meandering in laminar inflow. Varying the

reduced frequency of platform motion can generate different coherent structures in the wake, with some regimes dominated by surge-induced effects and others by meandering behaviour. These wake mechanisms, however, remain strongly influenced by the turbulent characteristics of the inflow. Inflow turbulence can affect wake evolution by limiting oscillations induced by platform motion and by altering wake recovery. Notable studies addressing this aspect include the experimental campaign by Messmer et al. (2024b) and high-fidelity numerical investigations by Pagamonci et al. (2025) and Li et al. (2024). The

combined impact of inflow turbulence and platform motion on wake recovery, platform-induced wake coherent structures, and wake meandering therefore remains an open area of research.

    Altogether, these findings highlight how platform motion can influence wake dynamics and contribute to wake re-energization, ultimately affecting the power production of downstream turbines. Capturing these complex phenomena requires the use of high-fidelity modelling techniques particularly when analysing interactions between floating wind turbines.

Only in recent years wake interactions in floating wind turbines have become the focus of dedicated research. Within the field of Unsteady Reynolds-Averaged Navier–Stokes (URANS) computational fluid dynamics (CFD) simulations, Rezaeiha and Micallef (2021) employed an actuator disk model to study the effect of upstream turbine surge motion on its interaction with a downstream turbine. While this approach provides a computationally efficient tool to explore motion-induced wake behaviour, the simplification of the rotor as a disk significantly limits the ability to capture detailed wake dynamics, including

wake recovery and the dominant flow frequencies in the mid- and far-wake regions.

    To address these limitations, the use of an Actuator Line Model (ALM) is essential, as it enables the resolution of the vortex structures generated by the turbine discrete blades. In this context, Arabgolarcheh et al. (2023b) and Arabgolarcheh et al. (2023a) adopted an ALM coupled with URANS simulations to investigate wake interactions between two floating turbines in a tandem configuration. However, for a more accurate representation of inflow turbulence, motion-induced wake features,

and their impact on downstream turbine loading, higher-fidelity approaches such as Large Eddy Simulations (LES) are crucial.





Research on the interaction between floating turbines using LES remains limited in the literature. A notable contribution is the work of Li et al. (2025), where LES–ALM simulations were used to explore wake dynamics in tandem turbines subjected to various combinations of surge motion.

The present study aims to contribute to the existing literature by employing a high-fidelity LES–ALM approach to investigate the interaction between two wind turbines, specifically in scenarios where the upstream turbine undergoes platform motion. As a novel aspect of this work, the upstream turbine is subjected not only to surge motion but also to pitch motion, enabling a comparative analysis of how these different motions influence wake structures and, in turn, affect the loading on the downstream turbine. The first section aims at a detailed characterization of the wake generated by the single floating upstream turbine, focusing on mean wake deficit, turbulence intensity, wake fluctuations induced by platform motion, wake recovery, and wake meandering. Once the single wake has been characterized, a second turbine is introduced downstream, and its aerodynamic loading is analysed in terms of mean values and fluctuations.

In parallel, this study enables a dual comparison across multi-fidelity numerical approaches. The first comparison is conducted between URANS and LES with turbulent inflow, in order to assess the ability of URANS to capture the complex wake dynamics and the resulting loads on the downstream turbine. The second comparison involves LES calculations with laminar inflow versus LES with turbulent inflow, aimed at isolating and characterizing the effect of inflow turbulence intensity (TI) on wake evolution. This set of multi-fidelity simulations provides valuable insight into the origin of wake-related mechanisms, allowing for a clear distinction between the effects of platform motion and those induced by inflow turbulence on key wake phenomena such as recovery, dynamic structures, and meandering.

To assess the reliability of the results, the numerical simulations were designed to replicate an experimental campaign, enabling validation of the wake generated by the single floating wind turbine. The reference experiments were conducted in the wind tunnel at Politecnico di Milano within the framework of the NETTUNO research project ((NETTUNO Research Project, 2023) (Fontanella et al., 2024)).

The work is structured as follows. Section 2 outlines the case study, providing key information about the turbine geometry as well as the tested and simulated cases. Section 3 introduces the numerical model with subsections covering the ALM code structure, the CFD setup and the numerical approach. Section 4 presents the validation of single floating turbine loads against experimental measurements. Section 5 delivers a comprehensive analysis of the wake shed by the single floating turbine, addressing mean velocity profiles, turbulence intensity, wake recovery, the propagation of wake-induced oscillations, and wake meandering. Section 6 discusses the results of the double turbine configuration, with emphasis on the loads acting on both turbines, considering mean values, platform motion–induced oscillations, and overall variations. Concluding considerations are drawn in the last chapter on the comparison of the different numerical models adopted and their implications for the estimation of interactions between turbines.



## 2 Case Study

### 2.1 Turbine and test facility

Numerical simulations were set up to replicate an experimental campaign to validate the loads and wake characteristics of the
single floating wind turbine. Specifically, the experimental data were obtained within the framework of NETTUNO project,
carried out in the large-scale wind tunnel at Politecnico di Milano (NETTUNO Research Project (2023), Fontanella et al.
(2024)). The experimental wind turbine is a 1:75 downscaled model of the DTU 10-MW reference turbine (Bak et al., 2013),
with its relevant geometric features listed in Table 1. The rotor blades were specifically designed to achieve similar thrust coef-
ficient of the full-scale DTU 10-MW turbine. The blades employ the low-Reynolds number airfoil SD7032, with modifications
to chord length and twist profile to reach the targeted thrust coefficient (Bayati et al., 2017). Wind speed was scaled by a factor
of 1:3, while preserving the original tip-speed ratio of the full-size machine. During the tests the tower was tilted by 5° ensuring
that the rotor plane was perpendicular to the incoming flow. The test turbine was mounted on a six-degree-of-freedom robotic
platform capable of replicating the rigid-body motions typical of floating wind turbine foundations.

**Table 1.** Geometry of the wind turbine used in the experiment (Fontanella et al., 2024)

| Parameter | NETTUNO value |
| --- | --- |
| Rotor diameter [m] | 2.381 |
| Blade length [m] | 1.102 |
| Hub diameter [m] | 0.178 |
| Rotor overhang [m] | 0.139 |
| Tilt angle [°] | 5 |
| Tower to shaft distance [m] | 0.064 |
| Tower length [m] | 1.400 |
| Tower base offset [m] | 0.730 |

The wind tunnel features a test section measuring 13.84 m in width and 3.84 m in height with an overall length of 35 m,
resulting in a geometric blockage of about 8%. The experiments were conducted under a uniform free-stream velocity of 4 m/s,
with an inflow turbulence intensity of approximately 1.5%, and an air density of $\rho = 1.18 \, \text{kg/m}^3$

### 2.2 Load cases

This study presents a series of simulations in different operational scenarios, including both fixed-bottom and floating wind
turbine configurations, to validate the numerical results. Three primary cases are considered: a fixed-bottom reference, a plat-
form undergoing surge motion, and a platform undergoing pitch motion. All experiments and simulations were carried out with





a constant rotor angular speed of $\Omega = 240\,rpm$, representative of rated operating conditions, corresponding to a tip-speed ratio $TSR = \frac{\Omega R}{U_0} = 7.5$, with a wind speed $U_0 = 4\,m/s$.

The time-varying surge displacement $x$ and pitch angle $\theta_p$ are defined in Equations 1 and 2.

$$x(t) = A_s \sin\big(2\pi f_s t + \phi_s\big) \tag{1}$$

$$\theta_p(t) = A_p \sin\big(2\pi f_p t + \phi_p\big) \tag{2}$$

where $A_{s/p}$ represent the amplitudes, and $f_{s/p}$ the frequencies of the surge and pitch motions, respectively. The phase shift $\phi$ accounts for the motion lag and, in the cases considered, is set to zero to initiate the platform motion in the leeward direction. The instantaneous velocity of the platform due to these prescribed motions is obtained by differentiating the displacement expressions. A key aspect of floating turbine dynamics is the apparent wind, which refers to the relative wind speed experienced by the rotor due to platform movement. The resulting variations in apparent wind speed at hub height reach a maximum excursion of $\pm 0.2$ m/s in both the surge and pitch cases, namely $\pm 5\%$ of the mean wind speed. The platform displacement and the corresponding apparent wind velocity are shown in Fig. 1 as a function of one platform motion. Details of the amplitudes and frequencies used for the imposed platform motions are summarized in Table 2.

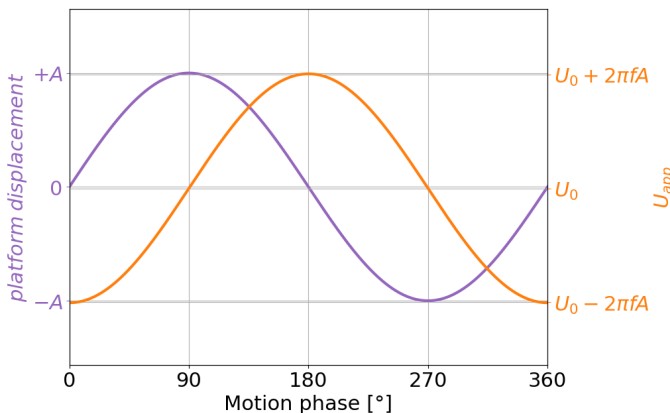

**Figure 1.** Platform displacement and apparent wind speed.

**Table 2.** Load cases

| Load case | $U_0$ [m/s] | $f\,[Hz]$ | $A\,[m]/[°]$ |
|---|---|---|---|
| Fixed-bottom | 4.0 | - | - |
| Surge | 4.0 | 1.0 | 0.032 |
| Pitch | 4.0 | 1.0 | 1.300 |



## 3 Computational approach

### 3.1 Actuator Line Model


All numerical simulations are carried out using an in-house actuator line model algorithm, developed and implemented within the OpenFOAM CFD framework, and previously validated in the URANS formulation in (Sanvito et al., 2024a). The simulation focuses exclusively on the three rotor blades, which are modelled as actuator lines comprising 75 actuator points where aerodynamic forces are applied. Figure 2 illustrates the flowchart of the methodology employed by the coupled CFD-ALM

solver.

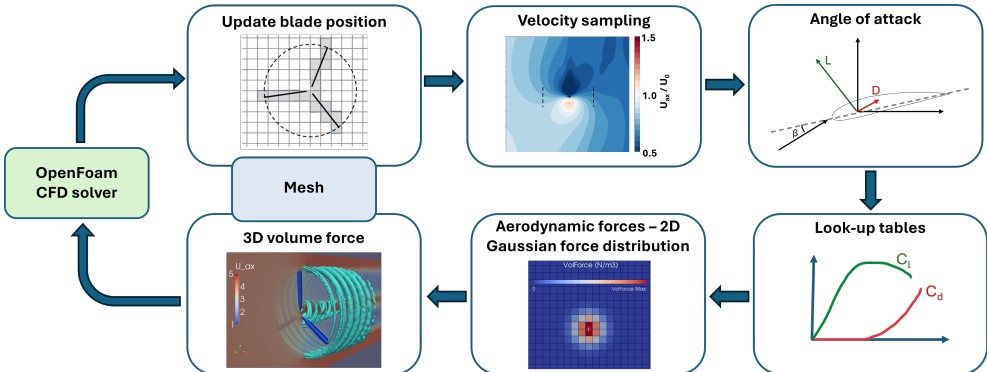

**Figure 2.** CFD-ALM algorithm flow chart.

The CFD module resolves the flow field, while rotor aerodynamics is modelled using Blade Element Momentum (BEM) theory. To accurately determine the axial and tangential components of the absolute velocity at each actuator point, this work adopts a custom velocity sampling method, called vortex-based sampling strategy and developed by the same authors (Sanvito et al., 2024a). Simple velocity extraction from the flow field is insufficient due to the influence of induction velocities caused

by the vorticity generated by the applied forces. To overcome this, the adopted methodology analytically calculates the induced velocity resulting from the bound vorticity associated with lift forces, using the Biot–Savart law. These induced components are subtracted from the sampled velocity to obtain a corrected velocity, which is then used to compute the angle of attack (AoA). The AoA is then used to compute the force coefficients from pre-tabulated aerodynamic polars, available for different Reynolds numbers and provided as input to the model. To ensure numerical stability, the point forces calculated at each actuator

point are distributed over the surrounding mesh cells using a 2D axisymmetric Gaussian smearing function. In this study, the regularization kernel has a width equal to twice the local rotor cell size. Given that the CFD-ALM approach resolves the blade tip vortices directly, no empirical correction models (such as Prandtl's tip loss correction) are required to account for tip-vortex-induced drag. However, the model does not inherently capture the reduction of aerodynamic forces due to pressure equalization near the blade tip. While such secondary effects can be addressed through ad-hoc corrections, as suggested in

Meyer Forsting et al. (2019), this aspect is not considered in the present work. Dynamic corrections of the aerodynamic polars





are also excluded from this study. A prior analysis (Bergua et al., 2023) showed that under the examined operating conditions, hysteresis effects are negligible since the AoA remains within the linear region of the polar curves.

Although the in-house ALM algorithm supports time-dependent control inputs (such as variable blade pitch and rotor speed) to simulate wind turbine control systems, these capabilities are disabled in this work, as all simulations are performed without

control and at a fixed rotational speed. The model configuration used here is consistent with that validated for similar load cases in Sanvito et al. (2024a).

To simulate the unsteady aerodynamics of floating offshore wind turbines, prescribed platform motions, as detailed in Table 2, are integrated into the CFD-ALM framework. The positions of actuator points are updated at each time step to reflect the instantaneous blade position, and their velocities are corrected accordingly using rigid body kinematics.

**3.2   Computational set-up**

The numerical domain is configured to replicate the geometry of the wind tunnel used in the experimental campaign. To account for the blockage caused by the boundary layer, the domain height is reduced by the boundary layer displacement thickness ensuring the same mass flow rate, and slip boundary conditions are applied to the wind tunnel walls, as done in Mancini et al. (2020). No alterations are made to the side walls, as their contribution to blockage is negligible; slip conditions are likewise

applied. At the outlet, atmospheric pressure is imposed, while the inlet is prescribed with a Dirichlet boundary condition corresponding to a uniform free-stream velocity for the URANS and laminar LES calculations. For the turbulent LES case, inflow turbulence is introduced using a synthetic turbulence generator to better reproduce experimental conditions. Specifically, the SynInflow tool (SynInflow, 2022) is employed, as it offers fine control over turbulence intensity and integral length scales, enabling accurate replication of experimental inflow characteristics while minimizing spurious pressure fluctuations. To coun-

teract the numerical decay of turbulence along the domain, the input parameters for SynInflow are carefully tuned to ensure that both turbulence intensity and energy spectrum match experimental turbulence statistics immediately upstream of the turbine location. A comparison between the simulated and experimental inflow turbulence is provided in Appendix A.

Due to the different character of the turbulent approaches considered, different mesh resolutions are employed; namely a coarser mesh for the URANS simulation and a refined mesh for the LES cases. For the URANS simulation, a structured

Cartesian mesh is adopted, consisting of approximately 20 million hexahedral cells. A grid sensitivity analysis, documented in Sanvito et al. (2024a), was conducted to verify mesh independence. In the rotor region, a characteristic cell size of $\Delta = 0.02\,m$ is assigned through the superposition of two cylindrical refinement zones. Figure 3 shows a zoomed view of the rotor area on the vertical plane passing by the rotor hub and the corresponding cell size. The turbulent LES simulation requires a higher resolution to ensure that at least 80% of the total turbulent kinetic energy (TKE) is resolved. To meet this criterion, a finer

background mesh is used, along with two cylindrical refinement zones. The second refinement zone, with $\Delta = 0.02\,m$, extends up to the domain inlet to mitigate numerical turbulence decay. The final mesh comprises approximately 113.2 million cells, with the most refined region having a characteristic cell size of $\Delta = 0.01\,m$, corresponding to a non-dimensional resolution of $\frac{R}{\Delta} = 120$. This level of refinement is consistent with values reported in the literature, such as Nilsson et al. (2015) ($\frac{R}{\Delta} = 122$), and Churchfield et al. (2017) and Blaylock et al. (2022), who employ $\frac{R}{\Delta} > 100$. For the laminar LES simulation, the same mesh



as the turbulent LES case is used, with the extension of the third refinement region ($\Delta = 0.01\,m$) to the inlet. This adjustment is introduced to suppress artificial numerical fluctuations at the inflow. As a result, the total cell count increases to approximately 150.9 million cells.

The URANS simulation uses the Realizable k-$\epsilon$ turbulence model, while the LES employs the Smagorinsky subgrid-scale model with a constant coefficient $C_s = 0.168$, suitable for the absence of no-slip walls. Both URANS and LES calculations use the PIMPLE algorithm for coupling pressure and velocity fields, as implemented in OpenFOAM. The algorithm is configured with two inner corrector loops to enhance convergence within each time step, and two outer loops. Regarding the numerical schemes, second-order Crank–Nicolson time integration is used, with second-order schemes for gradient, divergence, and Laplacian discretization.

The time step is chosen to limit the actuator line displacement to about one cell per step, ensuring numerical stability and accurate force application. Consequently, a time step of $\Delta t = 5 \cdot 10^{-4}\,s$ is employed throughout the simulations, resulting in a maximum Courant number of 0.54.

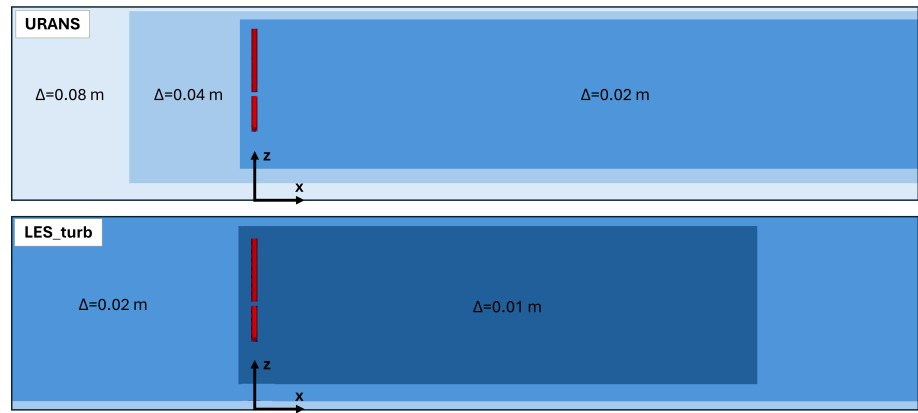

**Figure 3.** Zoom on the refinement around the rotor region for URANS and turbulent LES.

## 4 Single turbine rotor loads

At first the loads exerted by the machine in the single-turbine configuration across all simulated scenarios are analyzed. Table 3 provides a comparison between the three numerical simulations and the experimental data for the fixed-bottom case. In particular, it shows the average values along with the corresponding percentage error relative to the experimental data, according to the definition given in Equation 3.

$$err = \frac{mean(ALM) - mean(Exp)}{mean(Exp)} \tag{3}$$



Overall, the numerical results show good agreement with the experimental data, with a slight underestimation of thrust and an overestimation of torque. In any case, the numerical results are consistent with numerical counterpart indicating good numerical robustness of the ALM when combined with the different computational approaches. Regarding the floating cases,

**Table 3.** Fixed-bottom loads: comparison with experimental data.

|         | Thrust [N] | Torque [Nm] |
|---------|-----------|-------------|
| Exp     | 36.47     | 2.97        |
| URANS   | 35.58     | 3.16        |
| % err   | -2.43     | +6.31       |
| LES lam | 35.81     | 3.16        |
| % err   | -1.88     | +6.50       |
| LES turb| 35.87     | 3.18        |
| % err   | -1.65     | +6.95       |

200

Figure 4 shows the loads for the surge case (Fig. 4(a)) and the pitch case (Fig. 4(b)). Specifically, for each case the time evolution of the filtered thrust and torque over one platform oscillation period is presented, along with the percentage error on the mean value as defined in Eq. 3, and the percentage error on the oscillation amplitude, defined as in Eq. 4.

$$err = \frac{ampl(ALM) - ampl(Exp)}{mean(Exp)} \qquad (4)$$

To complete the analysis, Table 4 reports the mean thrust and torque values for surge and pitch.

**Table 4.** Surge and pitch mean loads.

|          | Surge | | Pitch | |
|----------|-----------|-------------|-----------|-------------|
|          | Thrust [N] | Torque [Nm] | Thrust [N] | Torque [Nm] |
| Exp      | 37.49     | 2.94        | 36.22     | 2.92        |
| URANS    | 35.69     | 3.18        | 35.62     | 3.18        |
| LES lam  | 35.75     | 3.17        | -         | -           |
| LES turb | 35.79     | 3.17        | 35.79     | 3.18        |

205

As in the fixed-bottom case, the numerical simulations tend to slightly underestimate the thrust and overestimate the torque. However, the mean thrust and torque values obtained in the pitch case are in very good agreement with the ALM simulations reported in Pagamonci et al. (2025), employing the same motion parameters. Additionally, a variation in the mean values of the experimental data is observed between surge and pitch. In general, differences in the mean values can be attributed to the zero



blade-pitch recalibration performed during testing, as noted by Fontanella et al. (2024). Once again, the numerical simulations
show excellent consistency with each other in terms of average thrust and torque. As for the amplitude, a good agreement with
experimental data is achieved with errors lower than 3% for the surge thrust and torque and for the pitch thrust. The largest
discrepancies are found in the torque amplitude for the pitch case; however, the numerical results appear consistent with the
corresponding surge case, where the turbine experiences the same variation of apparent velocity at the hub.

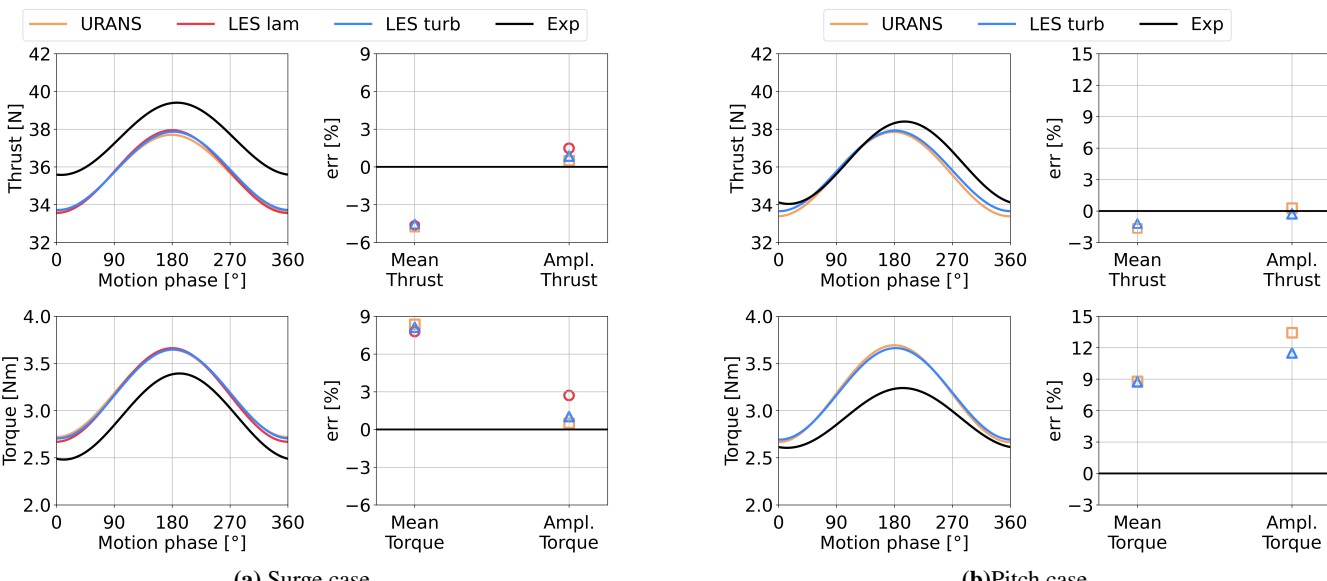

**(a)** Surge case.    **(b)** Pitch case.

**Figure 4.** Thrust and torque experimental validation in floating cases.

## 5   Single turbine wake analysis

This section presents a detailed analysis of the turbine wake in the single-turbine configuration across all case studies. The
objective is to provide an in-depth characterization of the wake that impinges on the downstream turbine, in order to better
interpret the resulting loads. The simulation dataset, including URANS, laminar LES, and turbulent LES for the fixed-bottom,
surge, and pitch cases, serves primarily to distinguish the effects of platform motion from those of the inflow turbulence on
wake behaviour, and secondarily to assess the accuracy of the numerical modelling approaches employed. The wake is first
validated with experimental data in terms of mean velocity profiles, turbulence intensity, and wake recovery. Subsequently, the
evolution of wake velocity fluctuations induced by platform motion is analysed to evaluate their impact on the downstream
turbine loading. Finally, wake meandering is investigated to determine whether its origin lies primarily in turbulent inflow
instabilities or is also significantly influenced by platform dynamics.

To this end, the following sections will analyse various flow quantities sampled along selected lines positioned in the wake.
Specifically, horizontal lines located at hub height and placed at downstream distances of 2D, 3D, 4D, and 5D will be used, as





well as vertical lines aligned with the rotor axis at 3D and 5D. The computational domain, actuator lines, and wake sampling lines are illustrated in Fig. 5.

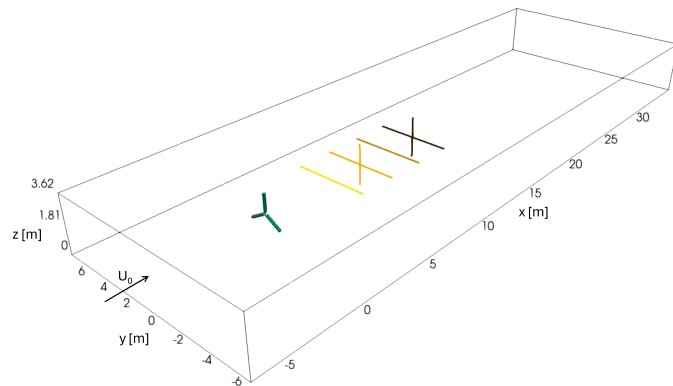

**Figure 5.** Numerical domain, actuator lines and sampling lines in the single turbine configuration.

## 5.1 Axial velocity and turbulence intensity: numerical vs. experimental comparison

Figure 6 shows the instantaneous axial velocity field, made dimensionless with respect to the freestream velocity, on a horizontal plane at hub height for the fixed-bottom and surge cases. In the fixed-bottom case (Fig. 6(a)), beyond the expected differences in wake structure resolution between URANS and LES, the comparison between the laminar LES and turbulent LES reveals that, despite the relatively low inflow turbulence intensity (around 1.5%), its impact on the wake is clearly visible. The wake in the turbulent LES case appears more chaotic, the tip vortices are less stable and mix out more rapidly.

In the surge case (Fig. 6(b)), coherent wake structures generated by the platform motion, corresponding to oscillations in axial velocity, can be identified. These structures are visible in the URANS simulation up to 6D downstream, before dissipating in the far wake. In the laminar LES, the same structures appear more pronounced, well-defined, and persistent, as the absence of inflow turbulence leads to a more stable and organized wake. Conversely, these structures are no longer distinguishable in the turbulent LES, where wake turbulence fluctuations dominate the instantaneous velocity field.

A more quantitative comparison of these aspects will be presented in the following sections. During the experimental campaign, wake measurements were carried out along downstream horizontal and vertical lines using a hot-wire anemometer with average uncertainties of $0.17\,m/s$ for flow speed and 0.4% for turbulence intensity (Fontanella et al., 2024). These measurements are used here to validate the numerical predictions of mean axial velocity and turbulence intensity profiles along horizontal lines at hub height. Figure 7 presents the validation for the fixed-bottom case for four different sampling lines.

The figure shows that the laminar LES underestimates the turbulence intensity in the wake, which limits wake recovery and results in greater discrepancies in the mean velocity profile. In contrast, both the URANS and turbulent LES calculations show better agreement with each other and with the experimental data in terms of mean velocity profiles. Regarding the turbulence





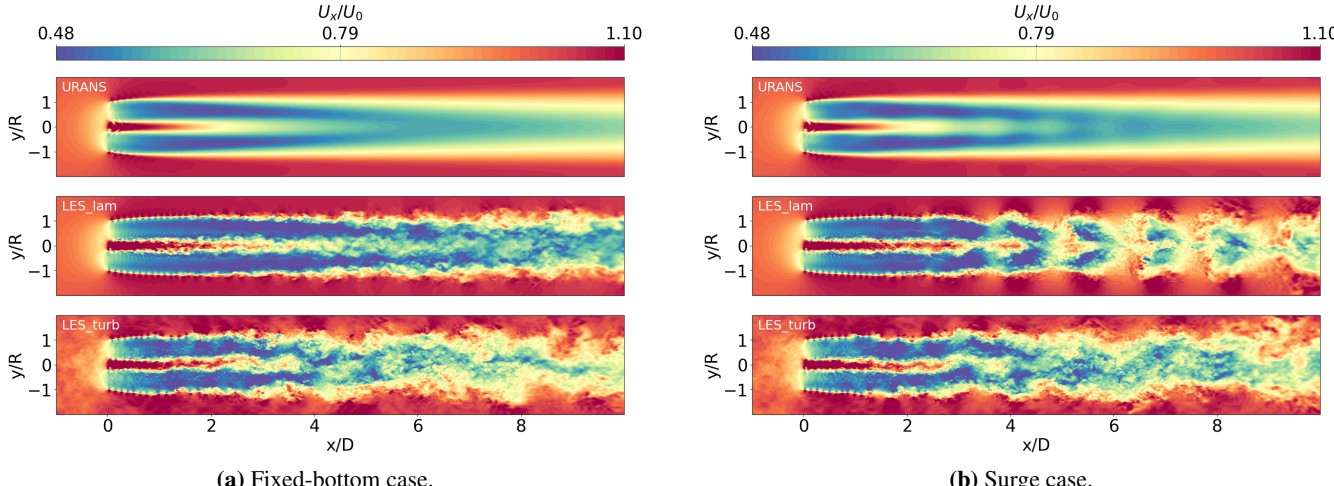

**(a)** Fixed-bottom case.  **(b)** Surge case.

**Figure 6.** Instantaneous axial velocity field on a horizontal plane.

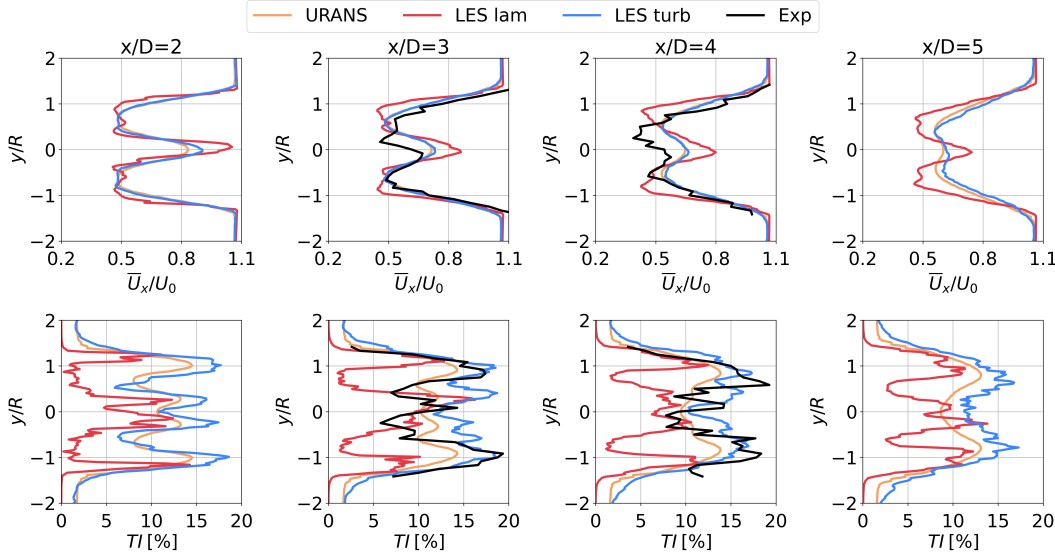

**Figure 7.** Mean axial velocity and turbulence intensity over four horizontal lines - fixed-bottom case.

intensity, although URANS provides reasonably accurate averaged values, the turbulent LES more accurately captures the local distribution, especially further downstream from the turbine.

Experimental validation was also performed for the surge and pitch cases. However, since the resulting observations do not differ substantially from those already discussed, the validation for these cases is provided in Appendix B for the sake of brevity.





To assess the consistency of the different numerical approaches, this analysis is extended by comparing the wake recovery process across the various cases. To better understand the impact of platform motion on wake recovery, the wake turbulence intensity trend is examined, as a key indicator of the phenomenon.

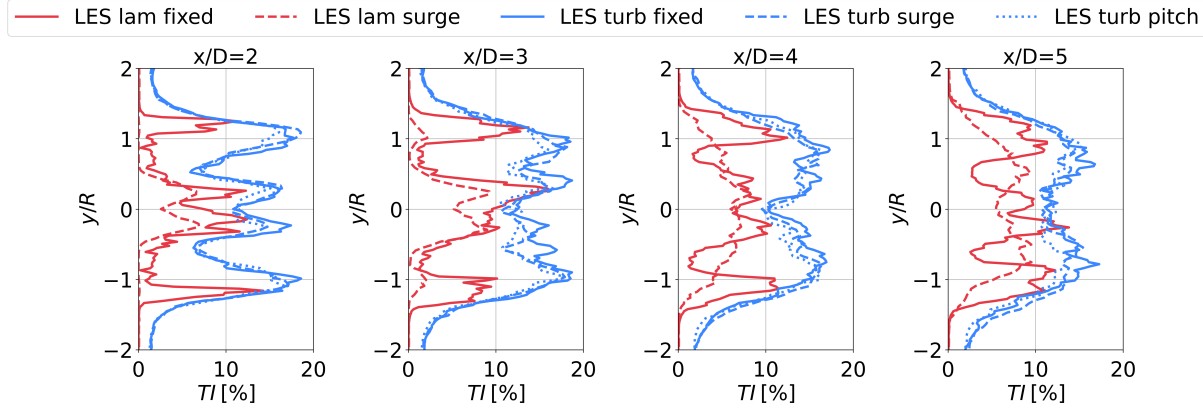

**Figure 8.** Turbulence intensity over four horizontal lines - LES calculations.


In particular, Figure 8 presents TI profiles along several horizontal lines, comparing laminar and turbulent LES cases to highlight their capability to resolve the largest turbulent structures. In addition to the laminar and turbulent fixed-bottom cases (solid lines), the laminar and turbulent surge cases (dashed lines) and the turbulent pitch case (dotted lines) are included. The comparison of the laminar cases is especially useful to isolate the contribution of added turbulence generated by rotor platform

motion. The figure indicates that, from 4D downstream onward, surge motion promotes a more uniform TI distribution, thereby enhancing turbulence intensity within the wake core. When turbulent inflow is introduced, however, the differences between the fixed-bottom and floating cases become less pronounced. This indicates that the increase in turbulence caused by platform motion, which is clearly non-negligible in laminar cases, almost vanishes even with a modest level of inflow turbulence. For the cases analysed, no significant difference is observed between the effects of pitch and surge motion under turbulent inflow

conditions.

To further support these observations, Figure 9 shows the mean axial velocity, computed using Equation 5 over each horizontal line, as a measure of wake recovery.

$$\frac{\langle \overline{U_x} \rangle}{U_0} = \frac{\sum_{i=1}^{n} \overline{U_x} |r_i|}{U_0 \sum_{i=1}^{n} |r_i|} \tag{5}$$

The velocity is first time-averaged at each point along the horizontal line and subsequently subjected to a weighted average

based on the distance of each point from the rotor centre (Eq. 5). The resulting trend aligns well with the turbulence intensity behaviour discussed above. In laminar inflow cases, surge motion leads to a noticeable increase in turbulence intensity, which enhances wake mixing and thus accelerates wake recovery. Under turbulent LES the wake recovery is already promoted by inflow turbulence so that the gain in the wake recovery capability in presence of surge motion is mitigated with respect to the




laminar cases. Again, the wake recovery observed in the pitch case is consistent with that of the surge case. These findings are in

line with previous results reported by Pagamonci et al. (2025) and Li et al. (2024), further supporting the physical reliability of the numerical simulations carried out in this study. Figure 10 shows the wake recovery comparison between turbulent LES and URANS simulations for the fixed-bottom, surge, and pitch cases. In the URANS results, a slower wake recovery is observed starting from 3D downstream, leading to differences in the estimated inflow velocity at a downstream turbine. Among the various URANS cases, the differences are negligible.

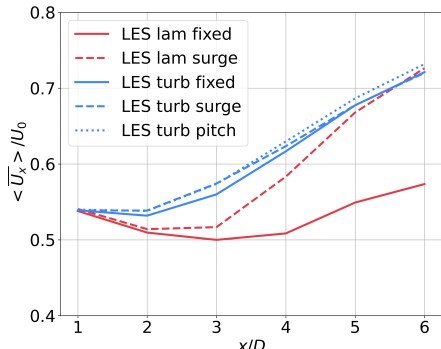

**Figure 9.** Comparison of wake recovery between laminar and turbulent LES.

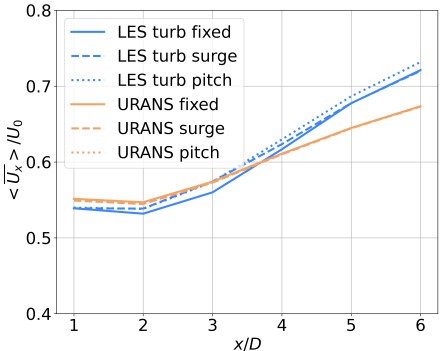

**Figure 10.** Comparison of wake recovery between turbulent LES and URANS.

**5.2 Platform motion induced wake velocity fluctuations**

As qualitatively illustrated in Fig. 6 (b), the platform motion induces velocity oscillations in the wake that propagate downstream and may influence subsequent turbines. The objective of this section is to quantify these oscillations in terms of phase shift relative to the platform motion and amplitude. To this end, only the results from the surge and pitch simulations are pre-





sented and compared. Figure 11 analyses the velocity fluctuations at a representative point in the wake, located at $y/R = -0.75$.

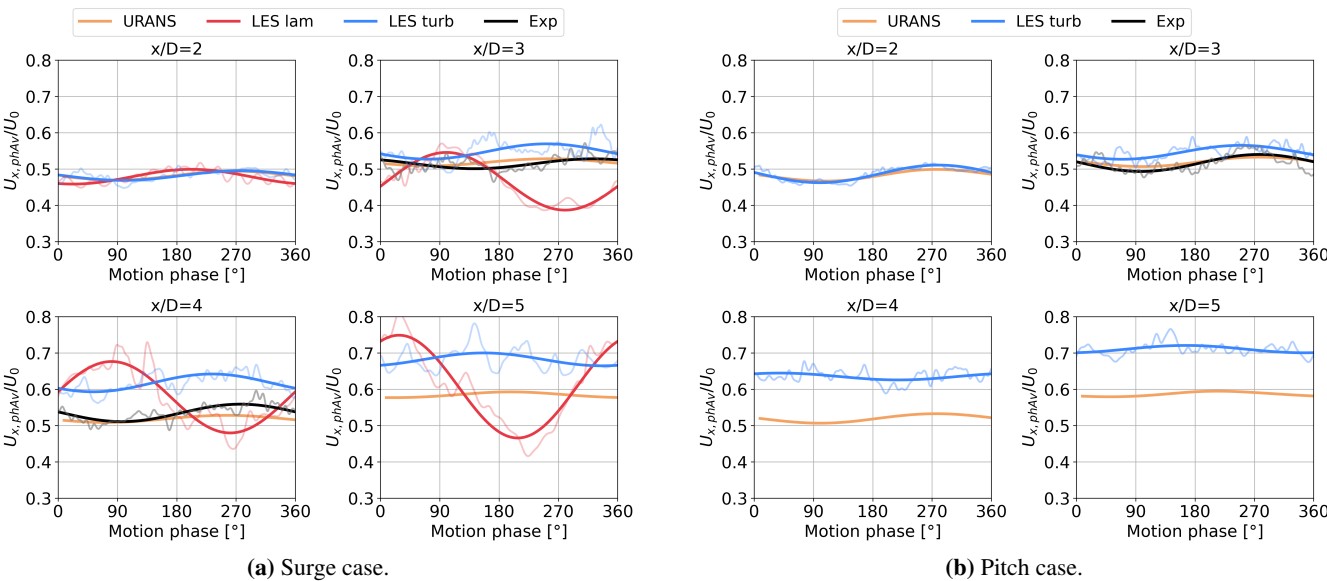

(a) Surge case.

(b) Pitch case.

**Figure 11.** Phase averaged stream-wise velocity over one platform period for a point at $y/R = -0.75$.

Specifically, the analysis is conducted at this same $y/R$ coordinate across various stream-wise positions $(x/D)$ downstream of the rotor. For each $x/D$ location, the stream-wise velocity signal is phase-averaged over a sufficient number of platform motion cycles to ensure a stable estimate of the trend, and it is shown as a function of the platform motion phase. For both the LES and experimental data, the original phase-averaged trends are shown together with the filtered trends to highlight the sinusoidal pattern at the same frequency as the platform motion. Figure 11(a) shows the results for the surge case. Focusing on

a fixed numerical model (e.g., turbulent LES), a noticeable phase shift emerges with increasing distance from the rotor due to wake propagation effects.

In terms of phase shift, the turbulent LES and URANS simulations show comparable behaviour at 2D, 3D, and 4D, with a slight deviation observed at 5D. This discrepancy may stem from different turbulence modelling approaches. Compared with the experimental data, a small phase difference is observed at 3D and 4D.

Regarding the difference in the mean value, it should be noted that the 75% blade span lies in a region of the wake characterized by high gradients, where estimates of mean velocity values are highly sensitive; nevertheless, the mean values remain consistent with those described in the previous section. It should be emphasized that this analysis involves a high level of detail; thus, the experimental data may be subject to sources of variability such as minor non-uniformities in inflow turbulence, imperfect control of platform motion or rotational speed. Nevertheless, the results are considered sufficiently reliable

and demonstrate good agreement between turbulent LES and URANS in terms of phase shift. Regarding amplitude, it can be qualitatively observed that URANS tends to underpredict the amplitude of the oscillations compared to turbulent LES and experimental data. A quantitative comparison of signal amplitudes will be presented later in the text. In contrast, the laminar





LES exhibits clear differences in both phase shift and amplitude. This highlights how the absence of inflow turbulence leads to excessively coherent and amplified wake structures that do not align with experimental observations. Figure 11(b) presents the

same analysis for the pitch case, where only the turbulent LES and URANS simulations are shown. Observations analogous to those previously discussed can be made here as well, indicating that, within the frequency and amplitude ranges considered in this study, the impact of pitch motion does not significantly differ from that of surge. Figure 12 presents a quantitative analysis

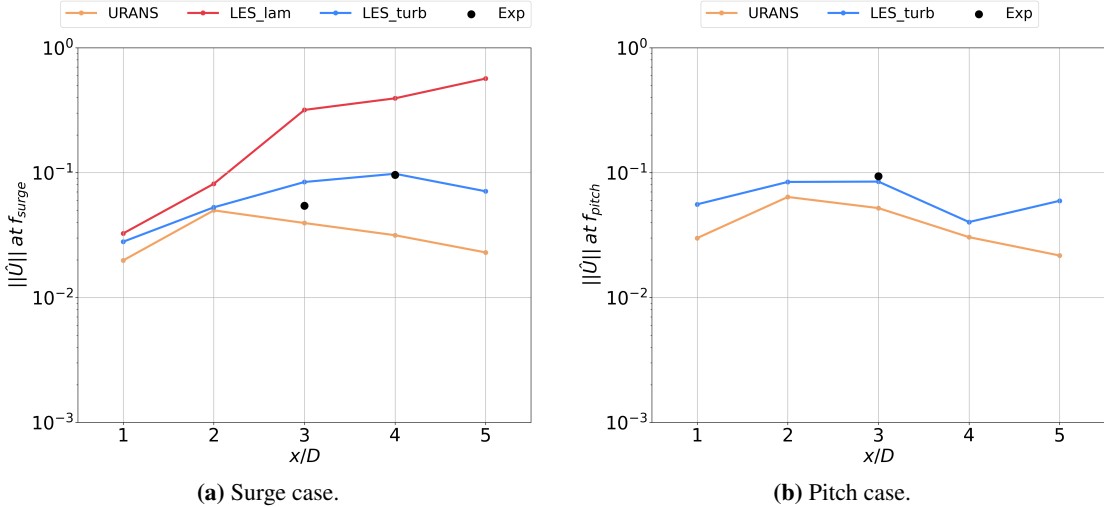

**(a)** Surge case.                                    **(b)** Pitch case.

**Figure 12.** Amplitude at the motion frequency for a point at $y/R = -0.75$.

of the amplitude, at the platform motion frequency, of the phase-averaged stream-wise velocity signal previously discussed. Specifically, it shows the amplitude of the phase-averaged stream-wise velocity at the point $y/R = -0.75$, computed as the

amplitude of the harmonic at the platform motion frequency using a Fourier Transform. In the surge case (Fig. 12(a)), the laminar LES demonstrates that the absence of inflow turbulence leads to an amplification of wake oscillation amplitude, which remains rather large even at 5D downstream of the rotor. After the downstream position at 2D, where wake fluctuations become more prominent, the URANS simulation increasingly underestimates the oscillation amplitude compared to the turbulent LES, likely due to the different turbulence model. The turbulent LES, on the other hand, shows good agreement with the experi-

mental data. For the pitch case (Fig. 12(b)), again no substantial differences are observed compared to surge, and the URANS simulation consistently underestimates the amplitude.

        The amplitude of wake oscillations is a key parameter for assessing whether a downstream turbine will experience unsteady inflow and how such fluctuations are spatially distributed across the rotor plane. To provide insight into the spatial distribution of wake oscillations at the position where the second turbine will be located (5D), Figure 13 displays the phase-averaged

stream-wise velocity over one platform motion cycle along a horizontal line at hub height. The figure includes both surge and pitch cases for the URANS and turbulent LES. For both pitch and surge motions, the highest oscillation amplitude is clearly concentrated in the central wake region, where oscillatory structures tend to persist longer compared to the outer shear layer. Turbulent LES calculations show a uniformly oscillating central region roughly between $y/R = -0.5$ and $y/R = 0.5$,




while URANS simulations exhibit symmetry between the two sides at $y/R = -0.5$ and $y/R = 0.5$. To quantify the amplitude of these oscillations, Figure 14 presents, for all four simulations, the amplitude of the harmonic component at the platform motion frequency for each point along the horizontal line. As anticipated qualitatively in the previous figure, both LES and URANS simulations confirm that the maximum oscillation amplitude occurs in the region between $y/R = -0.5$ and $y/R = 0.5$. Therefore, it is expected that the blade loads on the downstream turbine will be influenced by the platform motion primarily close to the hub region.

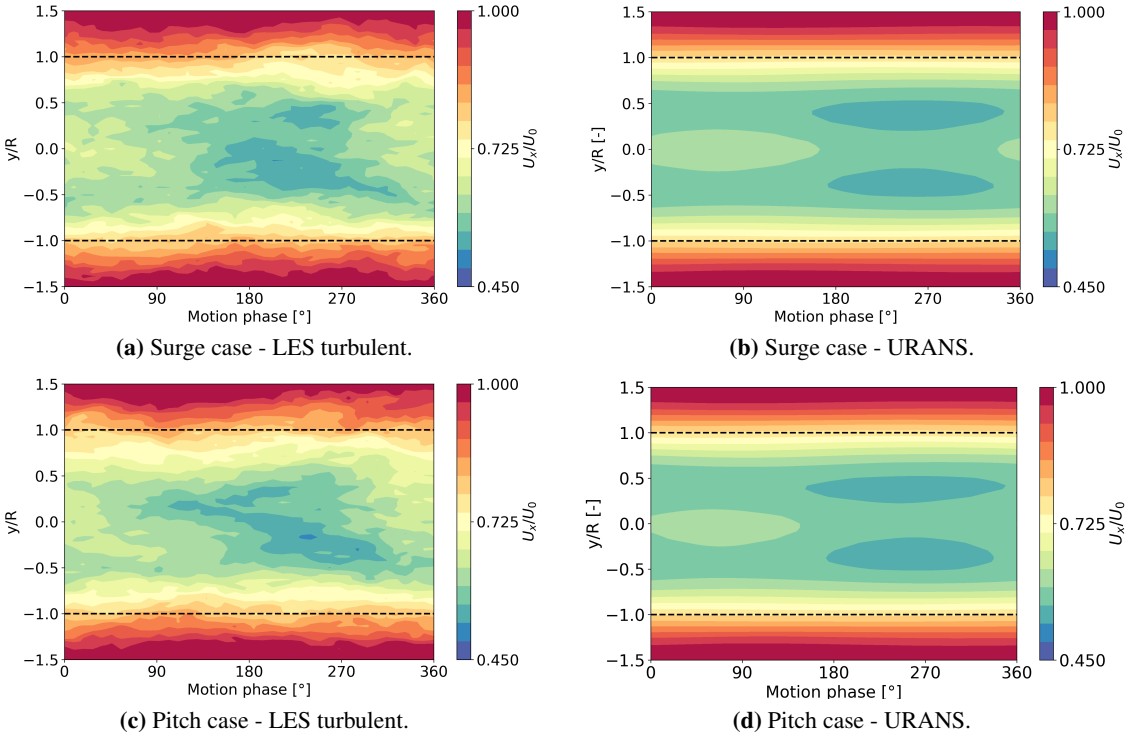

**Figure 13.** Phase averaged stream-wise velocity contours along a horizontal line at $x/D = 5$.

To conclude the analysis of wake oscillation propagation, the vertical plane aligned with the turbine hub is also examined to highlight any differences between the symmetric surge motion and the pitch motion, which introduces vertical velocity gradient in the wake. Figure 15 shows once again the phase-averaged stream-wise velocity, along a vertical line at $x/D = 3$. A comparison is made between all available numerical simulations, with the addition of experimental data. The figure refers to the surge case in the top row and the pitch case in the bottom row. In the surge case, the LES with laminar inflow visibly overestimates the oscillation amplitude, while the URANS simulation underestimates it. The numerical simulations generally display a symmetric behaviour between the upper and lower parts of the rotor, whereas the experimental data reveal a slight asymmetry. This asymmetry in the experiment is likely due to physical effects not included in the numerical simulations, such as the presence of the nacelle and tower. For the pitch case, no significant differences are observed compared to the surge case, either among the numerical simulations or in the experimental data. The pitch platform motion does not have a noticeable





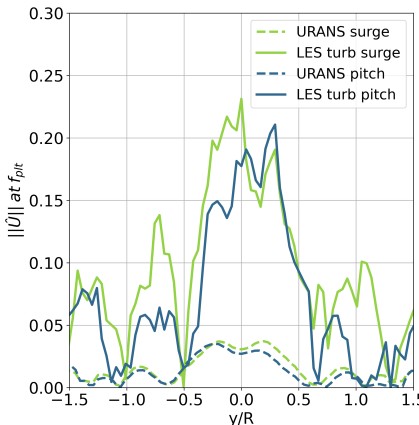

**Figure 14.** Amplitude at the motion frequency over a transverse horizontal line at $x/D = 5$.

impact on the wake, at least at a downstream distance of 3D. As in the surge case, the numerical simulations do not capture the asymmetry observed in the experiment. This experimental asymmetry is not attributed to a direct effect of the pitch motion, as it also appears in the surge configuration, suggesting it may arise from other experimental factors.

Figure 16 presents the same velocity fields along a vertical line at $x/D = 5$, with the aim of further characterizing the flow incoming into the downstream turbine. Consistent with the results in Fig. 14, the central region exhibits the largest oscillation amplitude in both the URANS and turbulent LES. In contrast, the laminar LES shows wider oscillations and less mixing, due to the absence of turbulent inflow. When comparing the two turbulent LES cases, a slight rightward shift of the minimum region in the pitch case is observed. Finally, the URANS simulations for both surge and pitch exhibit a slight phase shift in the oscillations compared to the turbulent LES results. In summary, the wake oscillations at 3D and 5D downstream are comparable between the surge and pitch cases, and the numerical simulations generally exhibit a symmetric wake behaviour. Minor differences between turbulent LES and URANS are mainly observed in terms of amplitude at 3D and phase shift at 5D.

### 5.3 Wake meandering

The last phenomenon analysed in the evolution of the wake of a single floating wind turbine is wake meandering, a process that can significantly affect the downstream turbine loads at low frequencies, thus being relevant for structural integrity. The availability of numerical simulations with varying levels of fidelity and different turbulence modelling approaches, both for fixed-bottom and floating configurations, allows for obtaining meaningful insights into the character of wake meandering under the considered operating conditions. The full analysis will focus on comparing the fixed-bottom case with the surge floating case, leveraging on the availability of the laminar LES. However, given the similar nature of surge and pitch motions, it is assumed that similar considerations will apply to the pitch case as well. The comparison between laminar LES calculations for the fixed-bottom and surge cases will help to isolate the effect of platform motion on wake meandering, independent of turbulent inflow conditions. The comparison between laminar and turbulent LES for the fixed-bottom configuration will



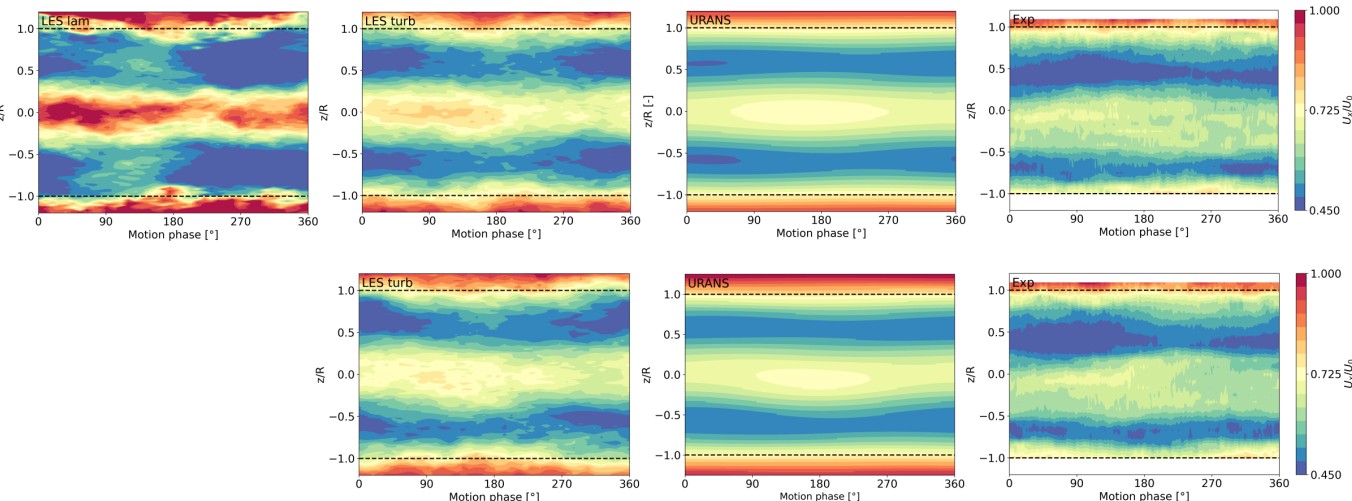

**Figure 15.** Phase-averaged stream-wise velocity contours on a vertical line at $x/D = 3$. First row: surge case. Second row: pitch case.

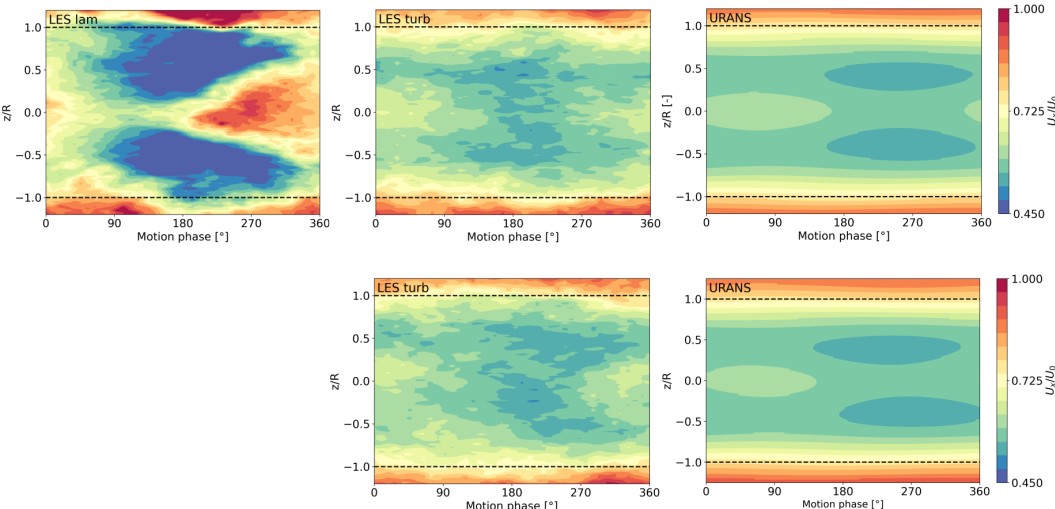

**Figure 16.** Phase-averaged stream-wise velocity contours on a vertical line at $x/D = 5$. First row: surge case. Second row: pitch case.

quantify the impact of turbulent inflow on the development of wake meandering. The comparison between turbulent LES calculations for the fixed-bottom and surge cases enables an assessment of the relative contributions of platform motion and turbulent inflow, as well as their combined effect on wake meandering under realistic turbulent inflow conditions. Finally, wake meandering will also be quantified for the URANS simulations, in order to evaluate the capability of such model to capture and reproduce the phenomenon.





In this section, the analysis will initially focus on the laminar and turbulent LES of the fixed-bottom and surge cases, in order to understand the fundamental characteristics of the phenomenon. Subsequently, the URANS simulations will be examined to complement the analysis and assess their predictive capabilities.

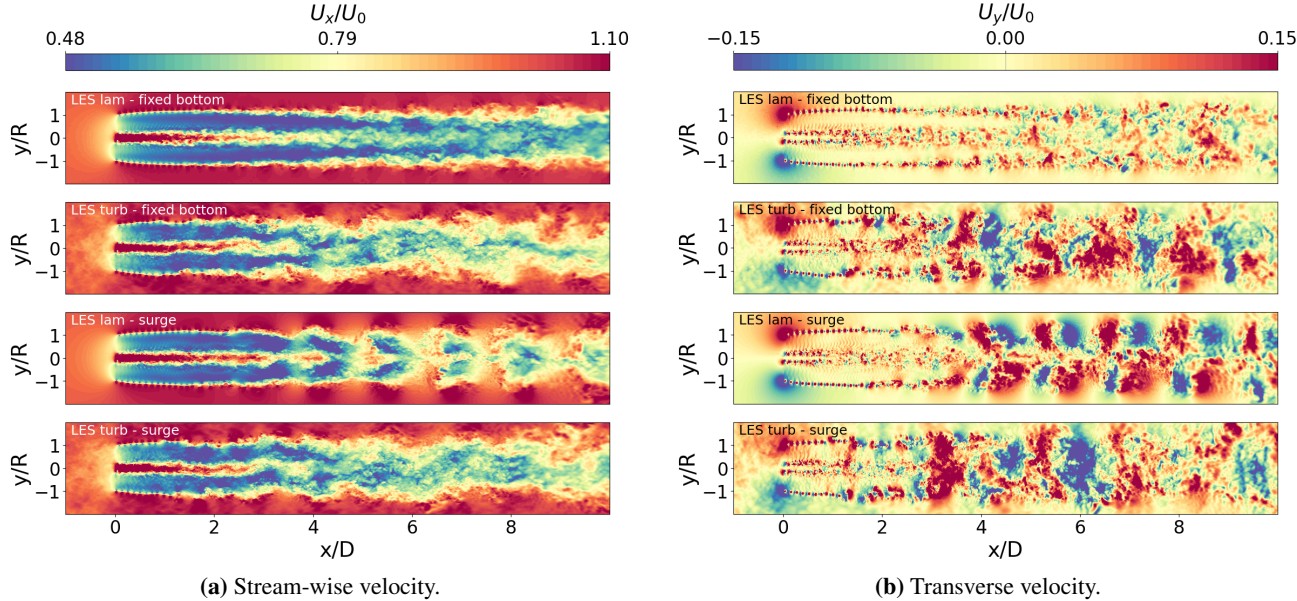

**(a)** Stream-wise velocity.

**(b)** Transverse velocity.

**Figure 17.** Instantaneous stream-wise and transverse velocity contours on a horizontal plane from various LES calculations.

The analysis focuses on the horizontal plane at hub height, where the larger domain extent minimizes the confinement effects
that take place in the vertical direction due to the wind tunnel limited height. Figure 17 presents the instantaneous stream-wise and transverse velocity fields on a horizontal plane for both laminar and turbulent LES calculations, in the fixed-bottom and surge configurations. Figure 17(a) shows that cases with turbulent inflow, regardless of platform motion, exhibit pronounced wake meandering, which is not observed in the laminar cases, with or without platform motion. To better characterize this behaviour, Figure 17(b) shows the transverse velocity field (component $U_y$), which is a useful indicator of wake orientation. In
the cases with evident wake meandering (turbulent fixed-bottom and turbulent surge), the $U_y$ field displays a distinct alternating pattern of positive and negative values from approximately 3D downstream, indicative of wake meandering. In contrast, the laminar fixed-bottom case does not display any distinct pattern. The laminar surge case, instead, exhibits a structured but qualitatively different pattern: opposite sign of $U_y$ across the traverse ($x/D$=const) with symmetry axis $y/R = 0$ suggests a pulsating pattern of the wake velocity rather than a periodic lateral displacement typical of meandering.
To further quantify the phenomenon and assess the influence of both surge motion and turbulent inflow, a spectral analysis is carried out to identify the characteristic frequency of the flow pattern caused by meandering. Figure 18 shows the power spectral density (PSD) of the transverse velocity for points along a horizontal line located 5D downstream of the rotor, for both laminar and turbulent simulations in the fixed-bottom and surge cases. The objective is to analyse the low-frequency spectrum to identify any dominant frequency contributions. In the laminar fixed-bottom case (Fig. 18(a)), no clear dominant frequency

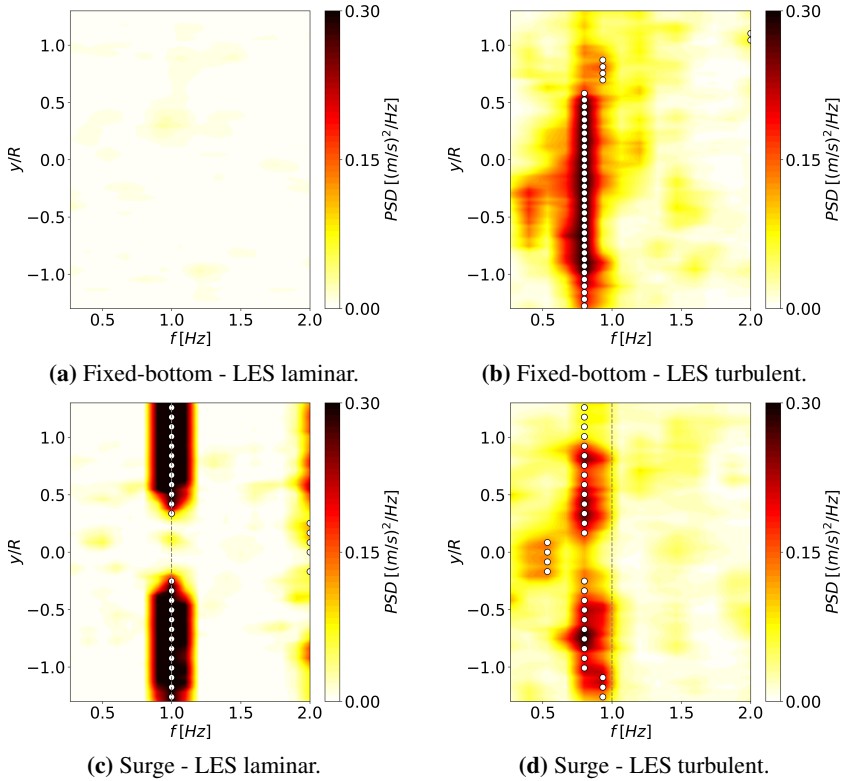

**Figure 18.** PSD spectra for the transverse velocity along a horizontal line at $x/D = 5$.

component is observed within the frequency range considered. In contrast, the turbulent fixed-bottom case (Fig. 18(b)) exhibits a dominant frequency in the range of $0.7-0.9\,Hz$ at all points along the line, which is deemed to correspond to the characteristic wake pattern previously identified in Fig. 17(b). In the laminar surge case (Fig. 18(c)), a dominant frequency again appears at all points along the line; however, the frequency matches the one of the platform motion itself ($1\,Hz$). This indicates that in laminar conditions, the wake dynamics are primarily governed by the platform motion, and no low-frequency component is identified. This confirms that the $U_y$ pattern observed in the laminar surge case has a different nature, being driven solely by the imposed platform motion. Interestingly, in the turbulent surge case, the dominant frequency observed at all points is no longer the one of the platform motion. Instead, a lower frequency component emerges in the $0.7-0.9\,Hz$ range, consistent with the turbulent fixed-bottom case. This indicates that the dominant wake dynamics in both turbulent scenarios arise from the same physical mechanism and are characterized by the same frequency. Therefore, under the operating conditions analysed, platform motion does not appear to play a significant role in enhancing wake meandering. Instead, the phenomenon is primarily driven by the turbulent inflow, which, despite its low turbulence intensity ($TI = 1.5\%$), increases wake instability and promotes the onset of wake meandering. The range of dominant frequencies identified in the turbulent cases corresponds to Strouhal numbers $St = fD/U_0$ between 0.42 and 0.53. These values are consistent with the typical range of [0.1–0.5] reported by Messmer





et al. (2024a) as characteristic of wake meandering induced by wake instabilities, particularly under low turbulence intensity
conditions ($TI < 5\%$), and further confirmed by Li et al. (2022) (Li et al., 2022).

To quantify these contributions, the wake motion induced by meandering is estimated by tracking the instantaneous position
of the wake center over time. Considering the strong instabilities in the wake velocity field caused by the combined effects of
turbulence and platform-induced oscillations, the estimation of the wake centre position is performed on different horizontal
lines at hub height and is preceded by data processing aimed at improving robustness. The commonly used centre-of-mass
approach (Hodgson et al., 2023) was found to be sensitive to instantaneous wake asymmetries induced by turbulent fluctuations
and platform motion, which do not necessarily reflect the global displacement of the wake. Moreover, under the operating
conditions analysed, the overall wake displacement is relatively small, prompting the adoption of a more robust method that
is less affected by the instantaneous asymmetry of the velocity profile. First, the deterministic contribution due to platform
motion is removed. This is achieved by subtracting the phase-averaged velocity signal at the platform motion frequency from
the instantaneous signal, effectively filtering out the platform-induced oscillation and stabilizing the wake profile. To further
suppress instantaneous wake instabilities caused by turbulence, each stream-wise velocity profile is approximated at every
time step using a Gaussian fitting procedure. As an illustrative example of the Gaussian fitting method, Fig. 19 shows the
stream-wise velocity profile at a generic time instant along with the corresponding fitted Gaussian curve for the turbulent surge
case.

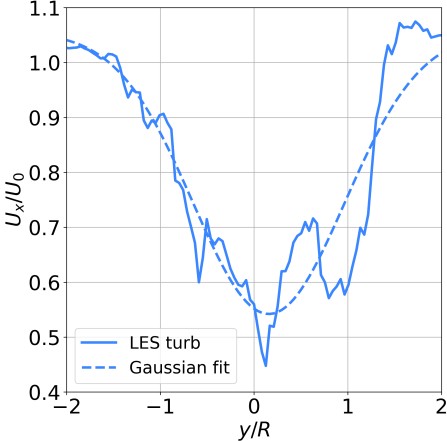

**Figure 19.** Instantaneous axial velocity profile and corresponding Gaussian fit at $x/D = 5$ for the turbulent surge case.

The wake center is therefore identified as the center of the Gaussian curve at each instant. The resulting time series of the
Gaussian center position represents the temporal evolution of the wake centerline and is used to characterize wake meandering.
This single-Gaussian approach is particularly well-suited for the far wake region, where wake meandering is most pronounced
and where the analysis is primarily focused.

To quantify wake meandering, the standard deviation of the wake centre position ($\sigma_Y$) is computed and shown in Fig. 20 for
all the LES cases analysed in this study. The laminar cases exhibit negligible meandering, confirming, as previously discussed,





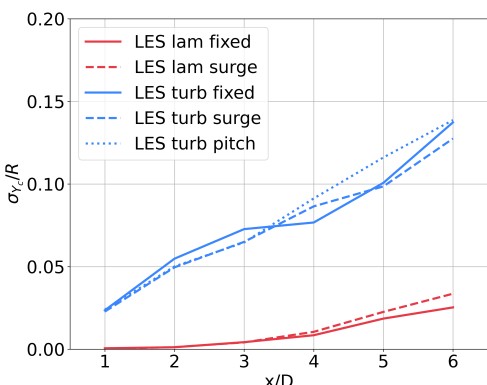

**Figure 20.** Standard deviation of the wake center position.

that surge motion does not trigger wake meandering in these working conditions. In contrast, the turbulent simulations display significantly higher meandering, as a result of the inflow turbulence-induced instabilities. The differences among the turbulent cases, fixed-bottom, surge, and pitch, are minimal, further confirming that, under the conditions examined, the effect of platform motion on wake meandering is not pronounced, either in surge or in pitch.

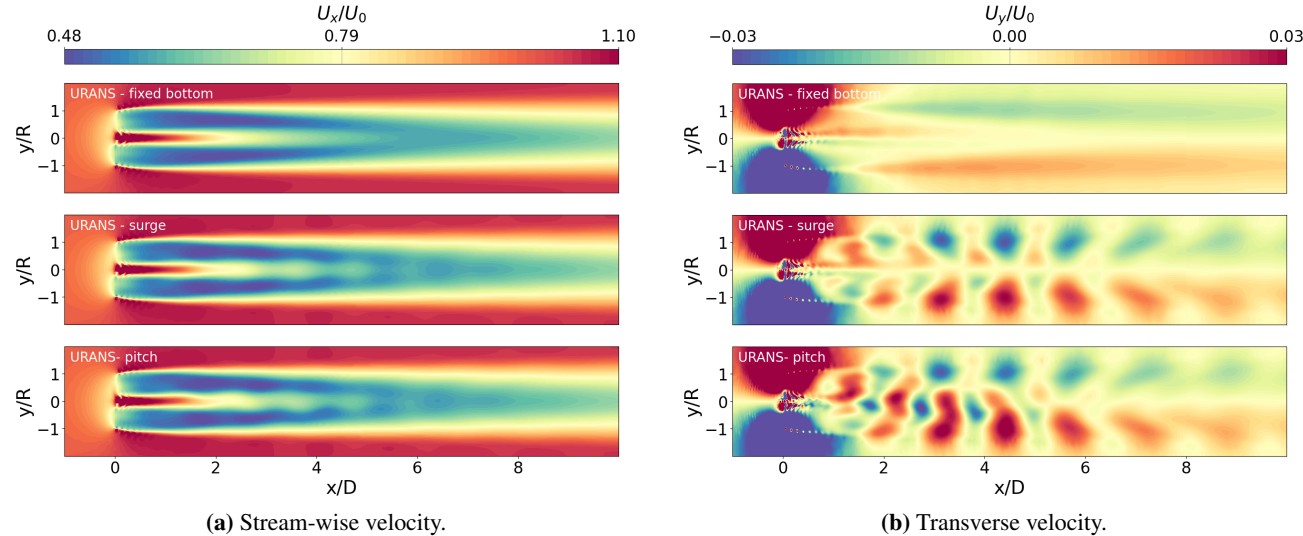

**(a)** Stream-wise velocity.                    **(b)** Transverse velocity.

**Figure 21.** Instantaneous stream-wise and transverse velocity contours on a horizontal plane from various URANS simulations.

With the origin of wake meandering established through turbulent LES calculations, the question arises whether URANS models are capable of capturing this phenomenon, which is inherently linked to turbulent flow instabilities. Figure 21 presents the axial and transverse velocity fields from the URANS simulations for the fixed-bottom, surge, and pitch cases on a horizontal plane. Figure 21(a) does not show any evident wake meandering. Consistently, the transverse velocity pattern in the surge





case of Fig. 21(b) resembles what observed in the laminar LES surge case, exhibiting a symmetric distribution of positive
and negative velocity regions that contributes to wake stabilization. In contrast, the pitch case in Fig. 21(b) displays slight
asymmetries up to 5D, though these remain insufficient to trigger wake meandering. As a result, the URANS simulations
reveal no displacement of the wake center.

This section has therefore examined the origin of the wake meandering captured in LES calculations and provided a quanti-
tative characterization of the phenomenon. Since the nature of wake meandering is strongly dependent on turbulent flow insta-
bilities, only LES calculations are able to capture it accurately. This highlights the importance of using high-fidelity approaches
when investigating wake dynamics that are tightly coupled with turbulence structures, as they are essential for accurately sim-
ulating and quantifying these phenomena and assessing their impact on downstream turbines in real atmospheric/turbulent
inflows.

## 6 Double turbine rotor loads

In this section, the objective is to analyse the impact of the previously characterized wake on the loads experienced by a
downstream wind turbine. To this end, a second turbine is introduced into the numerical domain, positioned 5D downstream
from the upstream turbine and aligned with it. The layout of the turbines within the computational domain is illustrated in
Fig. 22. From this point onward, the term WT will refer to wind turbine. The focus is specifically on two operating scenarios,
summarized in Table 5. In both cases, the upstream turbine undergoes surge and pitch platform motion, with the same amplitude
and frequency as the single-turbine configuration analysed in the previous chapters. The downstream turbine is kept in a fixed-
bottom position, allowing for the evaluation of load fluctuations induced solely by the oscillating wake of the upstream turbine.
The rotational speed of the upstream turbine is kept the same as in the previous cases, corresponding to a TSR of 7.5. The
rotational speed of the downstream turbine is set to 150 rpm, resulting in a TSR ranging between 7 and 8, depending on
whether the inflow velocity is calculated as the mean velocity sampled between 4D and 5D downstream of the upstream
turbine in the turbulent LES calculations with a single rotor.

**Table 5.** Load cases in the double turbine configuration.

|  | Upstream WT | | | Downstream WT | | |
|---|---|---|---|---|---|---|
|  | $\Omega$ [rpm] | $f$ [Hz] | $A$ [m]/[°] | $\Omega$ [rpm] | $f$ [Hz] | $A$ [m]/[°] |
| Surge | 240 | 1.0 | 0.032 | 150 | 0 | 0 |
| Pitch | 240 | 1.0 | 1.300 | 150 | 0 | 0 |

Figure 23 shows the instantaneous streamwise velocity fields on a horizontal plane at hub height for the double-turbine
configuration, for both surge (Fig. 23(a)) and pitch (Fig. 23(b)) cases. Once again, the comparison between the LES and the
URANS calculations highlights their different accuracy in capturing the flow field, particularly regarding the instabilities in the
wake of the downstream turbine.





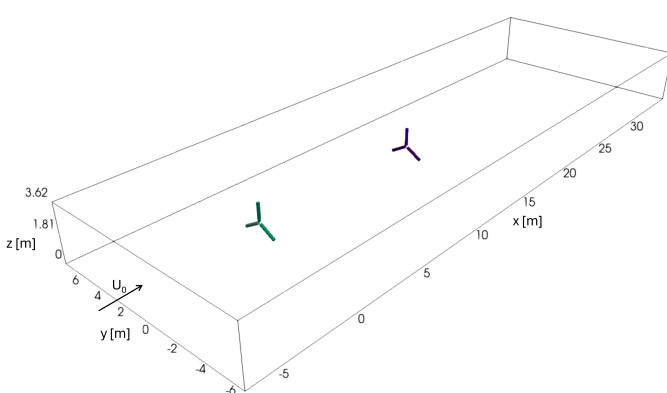

**Figure 22.** Numerical domain and turbines position in the multiple turbine configuration.

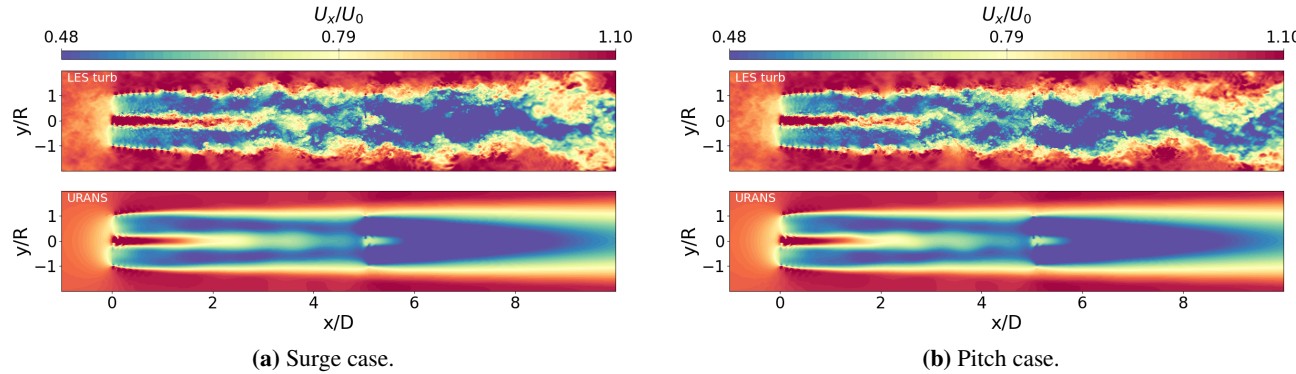

**(a)** Surge case.

**(b)** Pitch case.

**Figure 23.** Instantaneous axial velocity field on a horizontal plane in the double-turbine configuration.

Figure 24 shows the thrust force filtered at the platform motion frequency over one motion period, for both the upstream and downstream turbines. Specifically, it presents a comparison between the pitch and surge cases, as well as between turbulent LES and URANS simulations. For the upstream turbine, the thrust force exhibits a clear oscillatory behaviour dominated by the platform motion, consistent with the single-turbine configuration previously analysed (Fig. 4). The downstream turbine also exhibits an oscillatory thrust response, though with a markedly reduced amplitude. This behaviour aligns with the oscillations observed in the wake at a distance of 5D. Indeed, Figure 16 highlights a general phase shift: URANS tends to estimate the velocity trend with 90° phase shift after the LES prediction, and this phase lag is similarly reflected in the thrust signal. While URANS simulations predict nearly identical thrust responses for both surge and pitch motions, the LES results exhibit a slight phase difference between the two cases, which is also visible in Fig. 16. Consistent with the wake analysis, URANS generally underestimates the oscillation amplitude compared to LES.

To further investigate the local effects of platform motion on turbine loads, an analysis of span-wise quantities is presented. In particular, Figure 25 shows the radial distribution of the amplitude of the phase-averaged angle of attack at the frequency



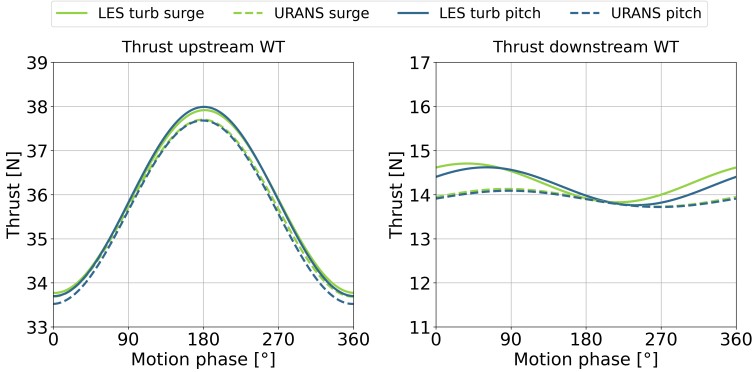

**Figure 24.** Thrust filtered at the platform motion frequency for the upstream and downstream turbines.

of the upstream turbine platform motion. To provide a comprehensive overview, the trend is reported for both the upstream and downstream turbines, under surge and pitch motions, using both turbulent LES and URANS simulations. To identify potential asymmetries induced by pitch motion or inherent in the wake, the results are presented for all three blades. For the upstream turbine, this representation highlights the AoA variations directly caused by the platform motion itself, and the comparison between blades helps detect any asymmetries. For the downstream turbine, the analysis reveals which rotor regions experience the greatest AoA fluctuations due to wake velocity oscillations, and the blade-to-blade comparison helps assess any non-uniformities in the wake impacting the downstream rotor.

Starting with the upstream turbine, both LES and URANS predict comparable AoA amplitudes across all radial positions, with slight differences near the root sections, likely due to different representations of the hub vorteces. In all plots, a larger AoA amplitude is observed in the root region, consistent with the fact that the lower peripheral speed makes the AoA more sensitive to variations in apparent wind speed induced by the platform motion. Furthermore, the comparison between the three blades reveals no significant asymmetries in the pitch motion case.

For the downstream turbine, larger AoA amplitudes are again observed in the root region, attributed to the same peripheral speed effect, in both LES and URANS calculations. However, in the LES results, these amplitudes are further amplified from the root up to about 50% span, due to stronger wake oscillations, as shown in Figure 14. The blade-to-blade comparison in the LES cases indicates a greater inflow non-uniformity in the root region. In the URANS simulations, the comparison reveals a slightly more visible asymmetry in the pitch case.

The analysis of the angle of attack is insightful, as this quantity is directly linked to wake velocity and allows for a verification of consistency with the findings discussed in the previous sections. However, the primary focus is the effect of platform motion on turbine loads. Accordingly, Fig. 26 presents the amplitude of the normal force for the same cases shown in Fig. 25. For the upstream turbine, no significant differences are observed between the surge and pitch cases when comparing the same numerical approach. The maximum amplitude predicted by URANS is slightly lower than that of LES, with the peak occurring at 65% of the blade span in the LES results and at 60% in the URANS results. For the downstream turbine, overall amplitudes





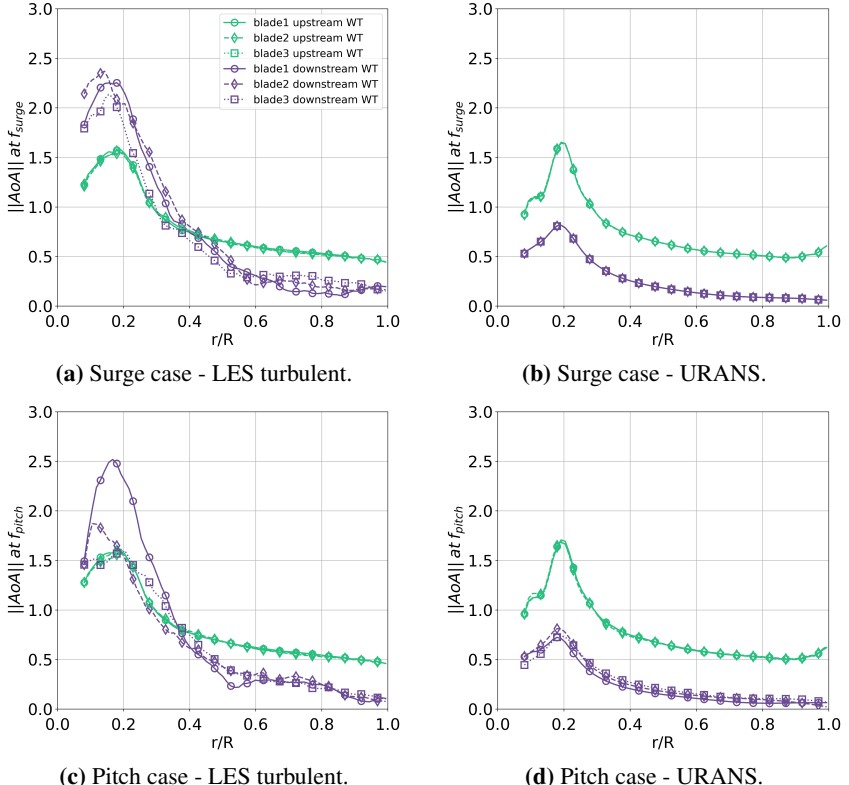

**Figure 25.** Spanwise attack angle amplitude at the motion frequency.

are lower along the entire span compared to the upstream turbine. In particular, URANS simulations show more limited
amplitudes, while LES results exhibit more pronounced blade-to-blade non-uniformities. Due to the spatial distribution of the
wake-induced oscillations impacting the downstream turbine, the maximum amplitude of the normal force occurs around 40%
of the blade span in both LES and URANS calculations.

The previous analysis has highlighted the extent of the variations of angle of attack and normal force primarily driven
by velocity fluctuations induced by platform motion. The focus now shifts to evaluating the combined effect of platform
motion and turbulence on these quantities. Figure 27 presents the mean value and standard deviation of the angle of attack for
both upstream and downstream turbines, under surge and pitch conditions, using LES and URANS calculations. In terms of
mean values, LES and URANS show good agreement for both operating conditions. For the upstream turbine, angle of attack
variations are predominantly driven by platform motion due to the low turbulence intensity of the inflow. As a result, URANS
and LES predict similar overall variability. In contrast, for the downstream turbine, LES results show significantly higher
variability, dominated by velocity fluctuations caused by the highly turbulent wake. URANS simulations, on the other hand,
only capture the deterministic contribution from the upstream platform motion and therefore predict much lower variability.
Once again, no notable differences are observed between surge and pitch cases when considering the same numerical approach.





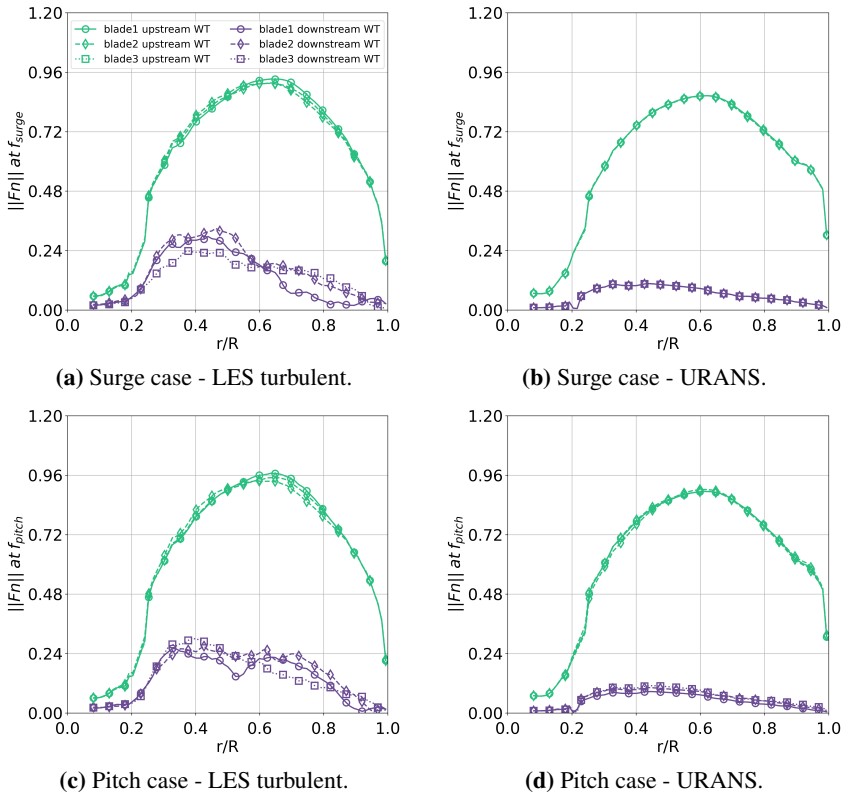

**(a)** Surge case - LES turbulent.

**(b)** Surge case - URANS.

**(c)** Pitch case - LES turbulent.

**(d)** Pitch case - URANS.

**Figure 26.** Spanwise normal force amplitude at the motion frequency.

A similar analysis is provided in Fig. 28 for the normal force to assess the variability of aerodynamic loads. Consistent
with previous observations, where variability is dominated by platform motion (upstream turbine), URANS captures similar
variations as LES. Where turbulence dominates (downstream turbine), URANS continues to exhibit only the deterministic
effect of the upstream platform motion. Therefore, this analysis has enabled a comprehensive characterization of the loads
acting on both turbines, both in global and local terms. URANS simulations clearly demonstrate a systematic underestimation
of load amplitudes on the downstream turbine due to wake-induced oscillations, as well as a significant under-prediction of the
overall load variability. Such an underestimation could severely compromise the accuracy of structural assessments, potentially
leading to reduced reliability in predicting the turbine long-term performance and fatigue life.

## 7 Conclusions

This work aims at providing insights into the wake dynamics of floating wind turbines and their impact on the loads experienced
by a downstream turbine. Numerical simulations with different levels of fidelity (URANS, laminar LES, and turbulent LES)
were carried out to isolate the effects of platform motion and turbulent inflow on wake behavior, as well as to assess the
ability of URANS to capture the wake characteristics of a floating turbine. A numerical Actuator Line Model was employed





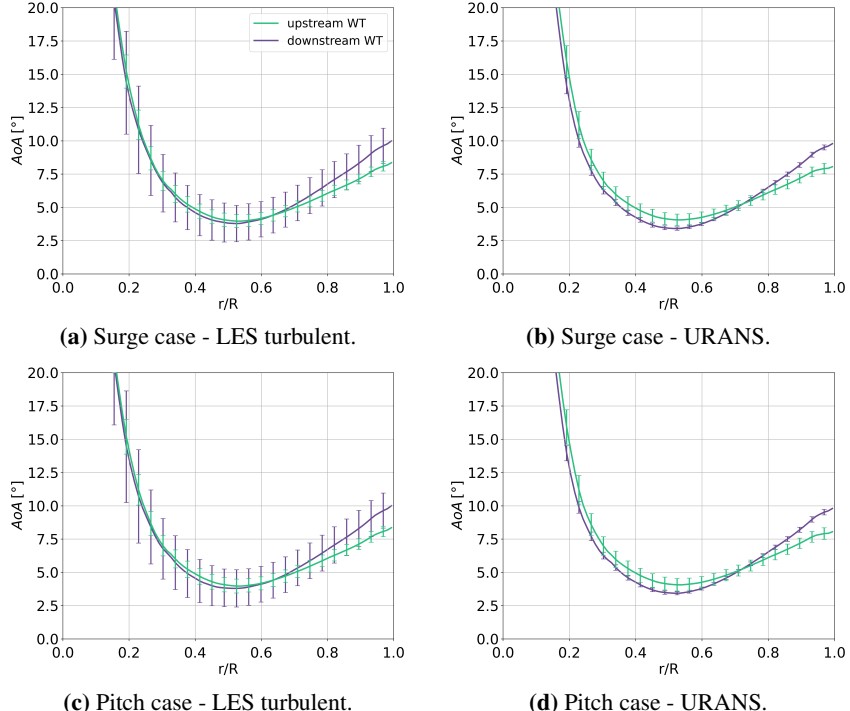

**(a)** Surge case - LES turbulent.

**(b)** Surge case - URANS.

**(c)** Pitch case - LES turbulent.

**(d)** Pitch case - URANS.

**Figure 27.** Spanwise mean and standard deviation of the attack angle.

to reduce the computational cost of rotor modeling while preserving the same accuracy level of CFD for the wake predictions. The analyzed cases included prescribed platform surge and pitch motions, in order to highlight potential differences in wake evolution and in the downstream turbine loads. The numerical setup was designed to reproduce accurately the corresponding

experimental campaign, enabling validation of both the loads and wake of the upstream floating turbine.

Validation of the upstream turbine loads under floating conditions showed good consistency across the different numerical approaches for both surge and pitch platform motions. In general, the simulations tended to slight underpredict thrust and over-predict torque compared to experiments even though absolute deviations remained limited. Good agreement with experimental data was achieved in terms of load amplitudes, particularly for thrust and torque in surge motion, and for thrust in pitch motion.

Wake validation, based on mean velocity profiles and turbulence intensity distributions, highlighted the essentiality of turbulent inflow in LES calculations and confirmed the superior ability of LES to reproduce the local distribution of turbulence intensity. Nevertheless, URANS proved to be capable of capturing mean velocity profiles consistent with both experiments and turbulent LES results especially in the near-wake. In terms of wake recovery, the numerical simulations showed good consistency between wake turbulence levels and the corresponding recovery rate. As already reported in the literature, the

wake-recovery benefit induced by platform motion is evident only under ideal laminar inflow conditions. Under turbulent inflow, the wake recovers more rapidly, and the beneficial effect of platform motion is significantly diminished. The comparison between turbulent LES and URANS highlighted a slower wake recovery in URANS, primarily in the far wake.


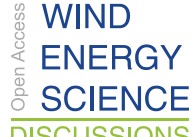

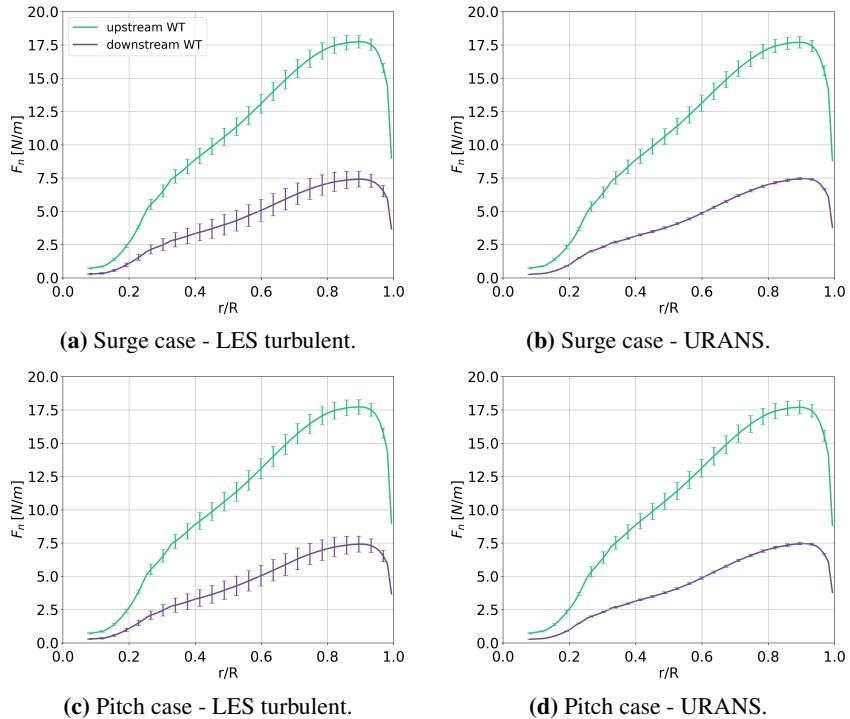

**(a)** Surge case - LES turbulent.

**(b)** Surge case - URANS.

**(c)** Pitch case - LES turbulent.

**(d)** Pitch case - URANS.

**Figure 28.** Span-wise mean and standard deviation of the normal force.

The analysis of wake oscillation propagation further highlighted the inadequacy of laminar inflow LES and revealed a slight phase-shift discrepancy between experimental data and both URANS and turbulent LES results. However, in terms of
oscillation amplitudes, turbulent LES provided estimates in good agreement with experiments, while URANS underestimated the effect of platform motion. No relevant differences were observed between surge and pitch cases, either in terms of phase shift or amplitude. The spatial distribution of wake oscillations indicated that both turbulent LES and URANS captured more pronounced oscillations at 5D downstream within the wake core ($y/R$ = -0.5 to $y/R$ = 0.5). Overall, the simulations exhibited fairly symmetric oscillations in the vertical plane, consistent with the perfectly symmetric inflow and turbine geometry. Once
again, URANS invariably underestimated oscillation amplitudes.

To complete the wake analysis, the phenomenon of wake meandering was examined both qualitatively and quantitatively to identify its origin. The availability of numerical simulations with varying fidelity levels and turbulence models, for both fixed-bottom and floating configurations, provided valuable insights into wake meandering under the studied conditions. The analysis compared laminar fixed-bottom and laminar surge floating cases to isolate the effect of platform motion from turbulent inflow.
The comparison between laminar and turbulent fixed-bottom configurations highlighted the influence of turbulent inflow on wake meandering, while turbulent LES comparisons between fixed-bottom and surge cases assessed the relative and combined effects of platform motion and turbulence. Specifically, turbulent LES calculations (with and without platform motion) revealed a wake meandering mechanism characterized by a distinct transverse velocity pattern and a characteristic frequency ($St \in$





[0.42, 0.53]). Quantitative analysis of the wake center displacement showed negligible wake meandering in laminar simulations, demonstrating that under the examined operating conditions, platform motion alone did not trigger the effect. Conversely, in turbulent simulations, the wake center displacement was comparable across fixed-bottom, surge, and pitch cases, confirming that wake meandering is primarily driven by turbulent inflow, which enhances wake instability and promotes its onset, while platform motion plays only a minor role. Given the turbulence-driven nature of the phenomenon, URANS simulations failed to capture wake meandering, resulting in negligible wake transversal displacement.

Finally, the impact of the characterized wake on the loads experienced by a downstream wind turbine was investigated. A second fixed-bottom turbine was placed 5D downstream in the wake of the upstream floating turbine. Loads on both turbines were first analysed in terms of overall thrust force. The propagation of wake oscillations influenced the downstream turbine loads, causing slight oscillations of thrust. Consistent with the previous wake analysis, URANS simulations underestimated the mean value and oscillation amplitudes and exhibited a slight phase shift compared to turbulent LES results. For a more detailed perspective, the radial distribution of angle of attack amplitude due to platform motion was examined. For the upstream turbine, AoA amplitude was dominated by the platform motion affecting the turbine itself; consequently, URANS and LES showed comparable amplitudes with no significant blade asymmetries. For the downstream turbine, both LES and URANS indicated increased AoA amplitude near the hub, partly due to lower peripheral velocity and partly due to larger oscillation amplitudes within the wake core. Regarding load amplitude, negligible differences were detected on the upstream turbine, whereas on the downstream turbine, URANS predicted smaller amplitudes and greater symmetry between blades. In terms of global variability of AoA and normal force, it was demonstrated that where variability is dominated by platform motion (upstream turbine), URANS captures variations similar to LES. However, where turbulence dominates (downstream turbine), URANS only reflects the deterministic effects of upstream platform motion, missing the contribution of high turbulence intensity.

In conclusion, this work provided a detailed characterization of the wake dynamics of a floating turbine, supported by the availability of experimental data. The diverse set of numerical simulations allowed to distinguish platform motion effects from turbulent inflow influence on wake features. The comparison between turbulent LES and URANS showed LES ability to reproduce local turbulence intensity distributions and to capture oscillation amplitudes induced by platform motion. URANS was found to be adequate in reproducing the mean wake velocity distribution in the near wake but it underestimates far wake recovery, oscillations and downstream turbine loads variability. URANS also failed to capture wake meandering ascribed to turbulent inflow instabilities, which is a prerogative of LES and high fidelity models. The findings demonstrate that employing LES enhances the accuracy in estimating loads on downstream turbines, which is essential for their structural assessment within wind farms.

*Code availability.* The ALM code has been developed in-house and, although it is implemented on an open-source platform, the modeling developments presented in this paper are not publicly available as open source.





*Data availability.* The experimental dataset is available at https://zenodo.org/records/13994980. Numerical data used in this study are available from the corresponding author upon request.

## Appendix A: Validation of the numerical synthetic turbulent inflow

The synthetic turbulent inflow is generated using the SynInflow tool (SynInflow, 2022), ensuring precise control over turbulence intensity and integral length scales. This approach enables an accurate replication of the experimental inflow characteristics

while minimizing spurious pressure fluctuations. To compensate for the numerical decay of turbulence along the domain, the input parameters for SynInflow are carefully calibrated so that both turbulence intensity and energy spectrum closely match experimental measurements immediately upstream of the turbine location. Figure A1(a) presents the vertical profile

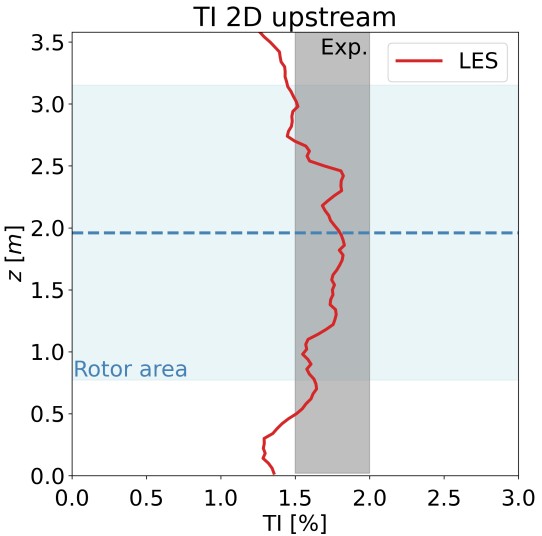
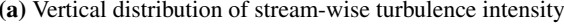

**(a)** Vertical distribution of stream-wise turbulence intensity

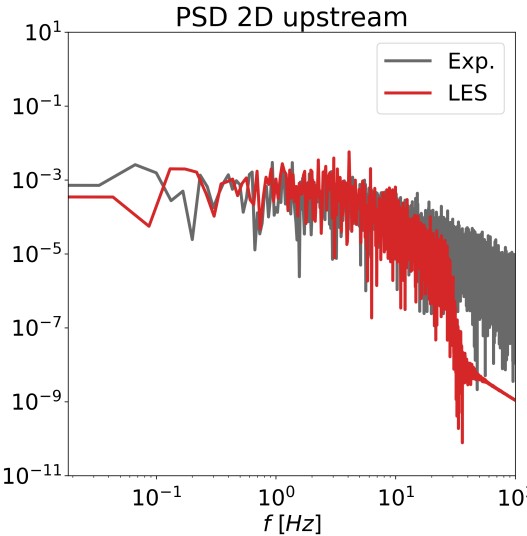

**(b)** Power spectrum density of stream-wise velocity

**Figure A1.** Turbulent inflow validation through numerical and experimental data comparison.

of turbulence intensity for the stream-wise velocity component, obtained from the numerical simulation at a distance of 2D upstream of the turbine. The numerical turbulence intensity distribution lies within the experimental range, highlighted in

grey, across the entire rotor area. Additionally, Figure A1(b) compares the power spectral density of the stream-wise velocity component from both numerical and experimental data in a point at the hub height and 2D upstream the turbine location. The two spectra exhibit good agreement up to the point where numerical LES filtering becomes noticeable.



**Appendix B: Wake experimental validation in surge and pitch cases.**

Figure B1 shows the dimensionless instantaneous axial velocity fields on a horizontal plane for the surge and pitch cases.
Overall, no significant qualitative differences are observed in the flow fields between the two conditions.

Figure B2 also presents the experimental validation of the surge and pitch cases in terms of mean velocity profiles and turbulence intensity along horizontal lines. In general, the laminar LES simulation underestimates the turbulence intensity in the wake, which limits wake recovery and leads to larger discrepancies in the mean velocity profile. In contrast, both the URANS and turbulent LES show better agreement with each other and with the experimental data in terms of mean velocity
profiles. As for turbulence intensity, while URANS provides reasonably accurate averaged values, the turbulent LES captures the local distribution more accurately, particularly further downstream of the turbine.

*Author contributions.* AF performed the simulations, post-processed the results, and contributed to data interpretation, visualization, and the original draft. AGS contributed to the development of the in-house code, study conceptualization, and data analysis. GP and VD supervised the project, contributed to its conceptualization, data interpretation, and manuscript preparation. All authors reviewed and contributed to the
final manuscript.

*Competing interests.* The contact author declares that none of the authors has any competing interests.

*Acknowledgements.* This study was carried out within the NEST - Network 4 Energy Sustainable Transition (D.D. 1243 02/08/2022, PE00000021) and received funding under the National Recovery and Resilience Plan (NRRP), Mission 4 Component 2 Investment 1.3, funded from the European Union - NextGenerationEU. This manuscript reflects only the authors' views and opinions, neither the European
Union nor the European Commission can be considered responsible for them.

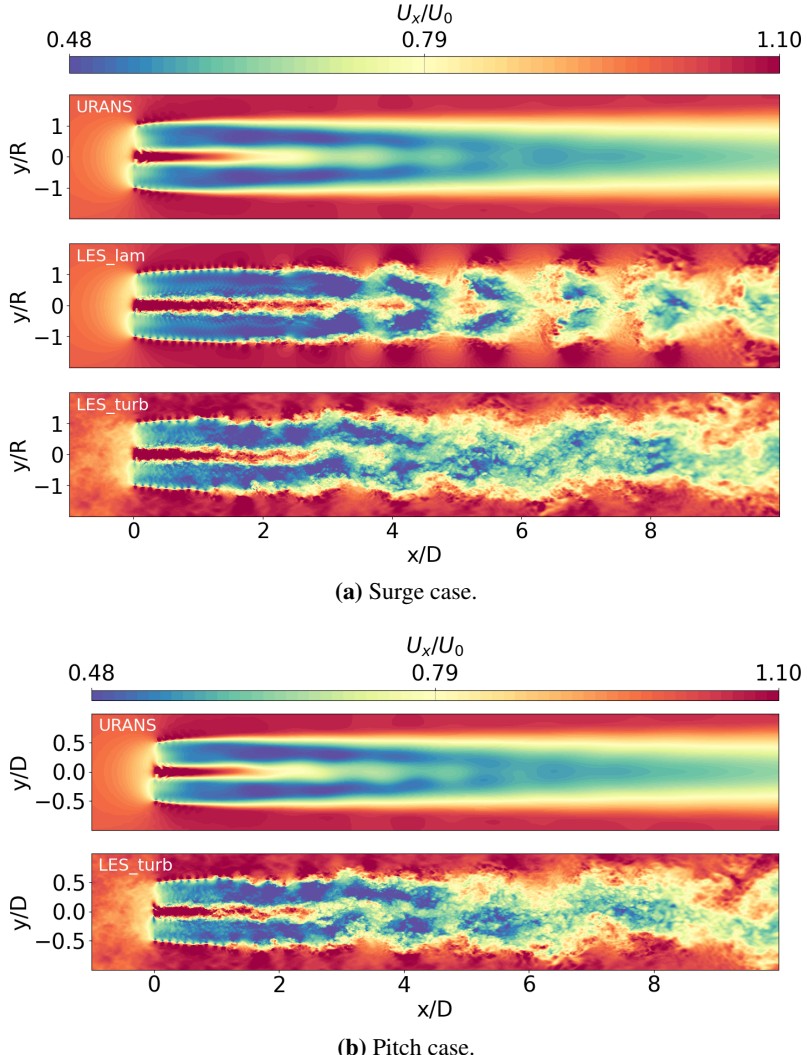

(a) Surge case.

(b) Pitch case.

**Figure B1.** Instantaneous axial velocity field on a horizontal plane.

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





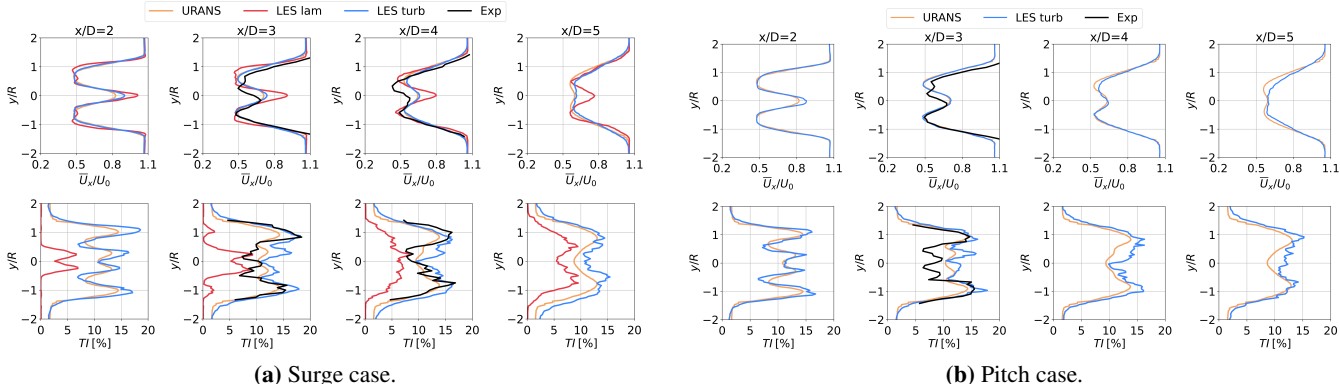

(a) Surge case.      (b) Pitch case.

**Figure B2.** Mean axial velocity and turbulence intensity over four horizontal lines.

Bayati, I., Belloli, M., Bernini, L., and Zasso, A.: Aerodynamic design methodology for wind tunnel tests of wind turbine rotors, Journal of Wind Engineering and Industrial Aerodynamics, 167, 217–227, https://doi.org/https://doi.org/10.1016/j.jweia.2017.05.004, 2017.

Berg, D. v. d., Tavernier, D. d., and Wingerden, J.-W. v.: Using The Helix Mixing Approach On Floating Offshore Wind Turbines, Journal of Physics: Conference Series, 2265, 042 011, https://doi.org/10.1088/1742-6596/2265/4/042011, 2022.

Bergua, R., Robertson, A., Jonkman, J., Branlard, E., Fontanella, A., Belloli, M., Schito, P., Zasso, A., Persico, G., Sanvito, A., Amet, E., Brun, C., Campaña Alonso, G., Martín-San-Román, R., Cai, R., Cai, J., Qian, Q., Maoshi, W., Beardsell, A., Pirrung, G., Ramos-García, N., Shi, W., Fu, J., Corniglion, R., Lovera, A., Galván, J., Nygaard, T. A., dos Santos, C. R., Gilbert, P., Joulin, P.-A., Blondel, F., Frickel,
E., Chen, P., Hu, Z., Boisard, R., Yilmazlar, K., Croce, A., Harnois, V., Zhang, L., Li, Y., Aristondo, A., Mendikoa Alonso, I., Mancini, S., Boorsma, K., Savenije, F., Marten, D., Soto-Valle, R., Schulz, C. W., Netzband, S., Bianchini, A., Papi, F., Cioni, S., Trubat, P., Alarcon, D., Molins, C., Cormier, M., Brüker, K., Lutz, T., Xiao, Q., Deng, Z., Haudin, F., and Goveas, A.: OC6 project Phase III: validation of the aerodynamic loading on a wind turbine rotor undergoing large motion caused by a floating support structure, Wind Energy Science, 8, 465–485, https://doi.org/10.5194/wes-8-465-2023, 2023.

Blaylock, M. L., Martinez-Tossas, L., Sakievich, P., Houchens, B. C., Cheung, L., Brown, K., Hsieh, A., Maniaci, D. C., and Churchfield, M. J.: Validation of actuator line and actuator disk models with filtered lifting line corrections implemented in Nalu-Wind large eddy simulations of the atmospheric boundary layer, in: AIAA SCITECH 2022 Forum, p. 1921, 2022.

Churchfield, M. J., Schreck, S. J., Martinez, L. A., Meneveau, C., and Spalart, P. R.: An advanced actuator line method for wind energy applications and beyond, in: 35th Wind Energy Symposium, p. 1998, 2017.

Fontanella, A., Fusetti, A., Cioni, S., Papi, F., Muggiasca, S., Persico, G., Dossena, V., Bianchini, A., and Belloli, M.: Wake Development in Floating Wind Turbines: New Insights and Open Dataset from Wind Tunnel Experiments, Wind Energy Science Discussions, 2024, 1–23, https://doi.org/10.5194/wes-2024-140, 2024.

Hodgson, E. L., Madsen, M. H. A., and Andersen, S. J.: Effects of turbulent inflow time scales on wind turbine wake behavior and recovery, Physics of Fluids, 35, 095 125, https://doi.org/10.1063/5.0162311, 2023.

Li, Y., Yu, W., and Sarlak, H.: Wake Structures and Performance of Wind Turbine Rotor With Harmonic Surging Motions Under Laminar and Turbulent Inflows, Wind Energy, 27, 1499–1525, https://doi.org/https://doi.org/10.1002/we.2949, 2024.





Li, Y., Yu, W., and Sarlak, H.: Wake interaction of dual surging FOWT rotors in tandem, Renewable Energy, 239, 122 062, https://doi.org/https://doi.org/10.1016/j.renene.2024.122062, 2025.

Li, Z., Dong, G., and Yang, X.: Onset of wake meandering for a floating offshore wind turbine under side-to-side motion, Journal of Fluid Mechanics, 934, A29, https://doi.org/10.1017/jfm.2021.1147, 2022.

Mancini, S., Boorsma, K., Caboni, M., Cormier, M., Lutz, T., Schito, P., and Zasso, A.: Characterization of the unsteady aerodynamic response of a floating offshore wind turbine to surge motion, Wind Energy Science, 5, 1713–1730, https://doi.org/10.5194/wes-5-1713-2020, 2020.

Messmer, T., Hölling, M., and Peinke, J.: Enhanced recovery caused by nonlinear dynamics in the wake of a floating offshore wind turbine, Journal of Fluid Mechanics, 984, A66, https://doi.org/10.1017/jfm.2024.175, 2024a.

Messmer, T., Peinke, J., and Hölling, M.: Wind tunnel investigation on the recovery and dynamics of the wake of a floating offshore wind turbine subjected to low inflow turbulence, Journal of Physics: Conference Series, 2767, 092 083, https://doi.org/10.1088/1742-6596/2767/9/092083, 2024b.

Meyer Forsting, A. R., Pirrung, G. R., and Ramos-García, N.: A vortex-based tip/smearing correction for the actuator line, Wind Energy Science, 4, 369–383, https://doi.org/10.5194/wes-4-369-2019, 2019.

NETTUNO Research Project: https://nettuno-project.it/, 2023.

Nilsson, K., Shen, W. Z., Sørensen, J. N., Breton, S.-P., and Ivanell, S.: Validation of the actuator line method using near wake measurements of the MEXICO rotor, Wind Energy, 18, 499–514, https://doi.org/https://doi.org/10.1002/we.1714, 2015.

Pagamonci, L., Papi, F., Cojocaru, G., Belloli, M., and Bianchini, A.: How does turbulence affect wake development in floating wind turbines? A critical assessment, Wind Energy Science Discussions, 2025, 1–36, https://doi.org/10.5194/wes-2024-169, 2025.

Rezaeiha, A. and Micallef, D.: Wake interactions of two tandem floating offshore wind turbines: CFD analysis using actuator disc model, Renewable Energy, 179, 859–876, https://doi.org/https://doi.org/10.1016/j.renene.2021.07.087, 2021.

Sanvito, A. G., Firpo, A., Schito, P., Dossena, V., Zasso, A., and Persico, G.: A novel vortex-based velocity sampling method for the actuator-line modeling of floating offshore wind turbines in windmill state, Renewable Energy, 231, 120 927, https://doi.org/https://doi.org/10.1016/j.renene.2024.120927, 2024a.

Sanvito, A. G., Firpo, A., Schito, P., Dossena, V., Zasso, A., and Persico, G.: Insights into the dynamic induction in FOWT surge motion using an actuator-line model, Journal of Physics: Conference Series, 2767, 052 064, https://doi.org/10.1088/1742-6596/2767/5/052064, 2024b.

Schulz, C. W., Netzband, S., Özinan, U., Cheng, P. W., and Abdel-Maksoud, M.: Wind turbine rotors in surge motion: new insights into unsteady aerodynamics of floating offshore wind turbines (FOWTs) from experiments and simulations, Wind Energy Science, 9, 665–695, https://doi.org/10.5194/wes-9-665-2024, 2024.

SynInflow: https://site.unibo.it/cwe-lamc/en/downloads/syninflow, 2022.

van den Berg, D., de Tavernier, D., and van Wingerden, J.-W.: The dynamic coupling between the pulse wake mixing strategy and floating wind turbines, Wind Energy Science, 8, 849–864, https://doi.org/10.5194/wes-8-849-2023, 2023.