# Peer review of "Multi-fidelity actuator line modelling of tandem floating offshore wind turbines"

_Wind Energy Science, 2025_

## Referee Comment (RC2)

**Review WES 2025-194**

November 25, 2025

**Introduction**

The paper by Firpo et al. investigates the wake of a floating turbine with imposed fore-aft motion (surge and pitch), using results from CFD simulation with two different methods (URANS and LES). The numerical data are compared with wind tunnel experiments from the Nettunto campaign. Overall, the paper is well written, the analysis is convincing, and the results are of interest to the community. So in principle suitable for a publication in WES journal. Please see my comments below.

**Key strengths of the article include:**

- **Comparison of laminar and turbulent backgrounds**: The analysis of wake recovery and dynamics for both fixed and fore-aft motion cases is convincing and aligns well with previous studies (e.g., Li et al. 2022, Messmer et al. 2024). Beyond this agreement, the paper's novel contributions—particularly the comparison of numerical methods and the insights into wake meandering—add value to the field.

- **Validation of LES against experimental data**: The comparison between simulation results and wind tunnel experiments is valuable.

- **Practical implications for downstream turbines**: The discussion on the potential impact on downstream turbines is of interest.

**Points for improvement:**

- **Limited scope of motion cases**: The study focuses on a single motion case—one frequency and amplitude of motion. While this choice is understandable for the purpose of comparing numerical methods (LES and URANS) with experimental data, it limits the generality of the conclusions regarding systematic differences between laminar and turbulent inflow conditions and their effects on wake recovery and dynamics.

*Suggestion*: The authors should explicitly acknowledge this limitation and justify the selection of this specific case. Additionally, they could frame the paper more clearly as a methodological comparison rather than a comprehensive analysis of motion effects.

*Additional cases*: To strengthen the findings, it would be valuable to include additional cases with larger amplitudes or varying frequencies, as turbulent inflow effects might differ significantly under these conditions.

- **Length and focus of the analysis**: The paper contains extensive analysis and numerous plots, which, may distract from the core contributions. The primary focus could be highlighted more effectively by streamlining the presentation.

**Additional comments:**

1. **Title Clarity**: As also noted by Referee 1, the current title is misleading. I recommend revisiting and refining it to better reflect the paper's focus and content, i.e remove the second part on impact on a downstream turbine and change 'multi-fidelity'.

2. **Literature Review**: The literature review could be expanded to include additional relevant studies, which you may have a look at, such as
Li, Yu, Sarlak in *Wind Energy* (2024)
Hubert, Conan, Aubrun in *Wind Energy Science* (2024)
Messmer, Peinke, Croce, Holling in *Journal of Fluid Mechanics* (2025)
Liu, Zhong, Zhao, Wan in *Physics of Fluids* (2025)
Mian, Messmer, Stoevesandt, Siddiqui in *Energy* (2025)

   which may provide further context or support for your results and analysis.

3. **Figure 4**: The plot shows a noticeable mean offset. Could the authors provide an explanation for this discrepancy? How precisely is the reference velocity $U_0$ determined in the experiments? While the dynamics appear well-captured, the offset remains unclear—why?

4. **Figure 10**: Please add the experimental point if you have (the figures 9 and 10 could be compiled)

5. **Figure 11**: Please clarify what constitutes "a sufficient number of platform motion cycles" for ensuring a stable trend estimate. Additionally, linking the platform motion signal with the phase of coherent fluctuations could provide deeper insight.

6. **Figure 12**: Please specify the unit used in this figure.

7. **Figure 12b (LES)**: The observation that the magnitude of pitch-related coherent fluctuations increases after 4D—despite decaying between 2D and 4D—is unexpected. What is happening there?

8. **Figure 14**: Please comment on the large differences between LES and URANS, especially knowing that in Mian, et al. *Energy* (2025), URANS was used to reconstruct flow structures, from my understanding, in a satisfying way. Here, it looks like the URANS is unable to reproduce the dynamics. Please add experimental points if available.

9. **Figures 15 and 16**: These could be merged into a single figure.

10. **Figure 17**: These results are of interest, particularly in Figure 17b. It is nice to see the difference between laminar and turbulent inflow in terms of wake meandering for surge. Focusing on the surge case (LES, turbulent inflow) around 2D, it looks like surge motion influences near-wake dynamics differently than fixed, and that the wake motions are a coupling between meandering and a mild plusating pattern. This could further be investigated and would add value to the paper. For instance, one can extend the analysis of figure 18 at different wake regions, for $x \in [1, 5]D$.

11. **Figure 18**: The meandering motion observed in the wake raises questions about the mechanisms triggering it. You could examine the free-stream power spectrum to see if the region around $f = 0.8$ Hz contains significant energy in the inflow, potentially linking free-stream structures to the wake dynamics.

12. **Figures 19 and 20**: The analysis on wake centre dynamics effectively demonstrates that surge and pitch motions do not significantly increase meandering with the turbulent inflow, nice result. These figures could be merged for conciseness.

13. **Figure 21 (URANS)**: The wake dynamics seem to retain some structural features of the motion, even though the inflow is turbulent, particularly the alternating blue and red regions of $U_y$, which suggest wake pulsating. Could the authors discuss this further, especially in relation to my previous comment 10.

14. **Downstream turbine**: The differences between LES and URANS on the loads should be linked to the results of the flow structures found in the wake, depending on the numerical method used.

Good luck with the revision,

---

## Author Comment (AC1)

**Rebuttal of reviews for paper: Multi-fidelity actuator line modelling of tandem floating offshore wind turbines**

The Authors would like to thank the Editor and the Reviewers for the evaluation of the draft version of the paper and for their relevant and constructive remarks and suggestions, which have considerably helped us in improving the paper through a significant revision.

In the following, the authors provide detailed answers and comments to all the remarks raised by the referees; the original reviewers' comments are shown in **black**, responses are shown in blue.

**Reviewer #1**

The manuscript "Multi-fidelity actuator line modelling of tandem floating offshore wind turbines" by Firpo et al. documents a numerical study on how flow simulation methods at different levels of fidelity (URANS, LES) impact the wake of a floating wind turbine modelled by actuator line methods. The impact of the wake on the loads of a second wind turbine placed downstream is also investigated. The paper confirms many known phenomena, such as that inflow turbulence reduces the difference between the wakes of a fixed and a floating wind turbine, and that LES is better than URANS when simulating wake dynamics. This manuscript could be considered for publication if the following comments are taken into account.

- My main concern is about the title of the manuscript. Conventionally, "multi-fidelity" modelling refers to simulation methods that employ models at different levels of fidelity simultaneously and consider information exchange at the interfaces between these models. However, this work compares URANS and LES without coupling them together in the same simulation. This cannot be referred to as multi-fidelity.
  We thank the reviewer for this observation. We understand the concern regarding the use of the term 'multi-fidelity' in the title. To avoid any confusion and for greater clarity, we have revised the title accordingly. The title now reads: "Actuator line URANS-to-LES comparison…"

- My second concern is also about the title. The focus of the work is mostly on the wake of a single floating wind turbine. It is only in Section 6 that the second wind turbine is introduced; only one of seven sections concerns the tandem configuration, so I don't think the title emphasizing tandem configuration is appropriate.
  We thank the reviewer, we understand the concern regarding the emphasis on the tandem configuration in the title. To better reflect the content and focus of the manuscript, we have revised the title accordingly. The title now reads "…comparison of single and tandem floating offshore wind turbines".

- The reason why URANS cannot reproduce large coherent structures should be studied in more depth. In principle, these large structures should not be so difficult to predict. Please verify whether the same phenomenon still occurs if the URANS employs the same grid as LES.
  In order to perform the URANS simulation on the same mesh used for the LES, the characteristic cell size would need to be halved, from Δ = 0.02 m to Δ = 0.01 m. Due to computational cost limitations, this analysis could not be carried out. However, the

reviewer's request to refine the mesh has been addressed: the surge-motion case was recomputed on a refined mesh with a characteristic wake cell size of 0.015 m, in order to exclude any potential influence of the spatial resolution on the prediction of the wake dynamics. Figure 1 shows the instantaneous non-dimensional freestream velocity field on a horizontal plane for both URANS simulations. From a qualitative point of view, no noticeable differences in the wake dynamics are observed.

[Figure]

Fig. 1

To provide a more quantitative assessment, Fig. 2 reports the amplitude of the motion-induced oscillations at a wake point located at y = 0.75 R. No significant differences emerge between the two meshes, indicating that the spatial resolution does not play a determining role in the estimation of the wake dynamics.

[Figure]

Fig. 2

The authors therefore believe that even adopting the LES mesh for a new URANS simulations would not eliminate the amplitude gap observed between the URANS and LES results, which suggests that the discrepancy may be related to differences in turbulence modelling approaches or, possibly, to numerical limitations inherent to the URANS solver.

In the revised version of the paper, a comment on this aspect has been added in Section 3.2. The comment has been included at lines 185-187 of the revised paper.

- It is found that the inflow triggers significant meandering in the wake. The spatial and temporal scales are very important to the receptivity of the wake. In Figure A1(b), the low-frequency peak of the experimental data and of the simulations differ considerably. Please try to see whether this discrepancy persists if a different inflow turbulence is employed.

  The authors agree with the reviewer that different spatial and temporal scales of the inflow can influence wake behaviour, as demonstrated for example by Hodgson, H. Aa Madsen, and Andersen ('Effects of turbulent inflow time scales on wind turbine wake behavior and recovery', Physics of Fluids, 35(9), 2023) and by Hodgson, Troldborg, and Andersen ('Impact of freestream turbulence integral length scale on wind farm flows and power generation', Renewable Energy, 238, 2025). However, this type of analysis is beyond the scope of the present paper, in which the focus is not on investigating the phenomenology of wake meandering (which may be influenced by different turbulent spectra or turbulence scales), but the capabilities of different computational models in predicting wake mixing and evolution. For this reason, the analysis concerning the identification and quantification of wake meandering is carried out exclusively through numerical simulations, keeping identical the turbulent inflow conditions for the LES cases, both in the fixed-bottom and surge, thus ensuring a fair comparison.

  Hence, the conclusions drawn on the exemplary cases studied refer specifically to the operating conditions investigated, for that the inflow turbulence was imposed by the wind tunnel facility, and are not intended to be generalized.

  Regarding the reviewer's specific comment on the mismatch between the low-frequency peak in the numerical and experimental spectra at the inflow, we note that the freestream spectral peak occurs at a lower frequency than the one identified as wake-meandering (0.8 Hz). Therefore, no evident link can be identified between the freestream frequency content and the wake-meandering frequency.

Some point comments are:

- Line 191, please specify how the Courant number is defined. If it is defined based on the flow velocity, then the rotor tip, traveling much faster than the local flow, could cross many grid cells in a single time step.

  As specified in the text, the timestep was defined to limit the actuator-line tip displacement to about one cell per step, ensuring numerical stability and accurate force application. The definition of the Courant number has been included in the revised paper for completeness, using the timestep formerly adopted (lines 204-207).

- Figure 3, it is suggested to add a subplot for the LES_laminar case as well.
  The figure has been added.

- The captions of the figures are not self-contained. It is not clear what the light-colored curves represent.
  We assume the reviewer is referring to Fig. 11. The legend has been updated to include all curves shown in the plot, and the caption has been extended to describe each curve

- Line 400, it is odd that a work published in 2024 is said to be confirmed by a work published in 2022.
  We thank the reviewer for pointing this out. The sentence has been revised to indicate that the values reported by Li et al. (2022) are in agreement with those reported by Messmer et al. (2024), rather than suggesting a chronological confirmation.

**Reviewer #2**

**Points for improvement:**

1. Limited scope of motion cases: The study focuses on a single motion case—one frequency and amplitude of motion. While this choice is understandable for the purpose of comparing numerical methods (LES and URANS) with experimental data, it limits the generality of the conclusions regarding systematic differences between laminar and turbulent inflow conditions and their effects on wake recovery and dynamics.

   *Suggestion:* The authors should explicitly acknowledge this limitation and justify the selection of this specific case. Additionally, they could frame the paper more clearly as a methodological comparison rather than a comprehensive analysis of motion effects.

   *Additional cases*: To strengthen the findings, it would be valuable to include additional cases with larger amplitudes or varying frequencies, as turbulent inflow effects might differ significantly under these conditions.
   Following the reviewer's suggestion, the Introduction (lines 66–87 of the new version of the paper) has been revised to clarify the scope and objective of the paper. The emphasis is now placed on the methodological comparison between different numerical models and their validation against experimental data, rather than on providing a comprehensive analysis of motion effects. Furthermore, it is explicitly stated that the conclusions regarding the impact of motion apply specifically to the operating conditions considered in this study. The same revisions have been applied to the Conclusions section to ensure consistency.

2. Length and focus of the analysis: The paper contains extensive analysis and numerous plots, which, may distract from the core contributions. The primary focus could be highlighted more effectively by streamlining the presentation.
   To address the reviewer's concern, some images have been combined into single figures to reduce the overall length of the manuscript. At the same time, the authors opted not to remove any analyses or results, in order to preserve the full scientific contribution of the paper. To further guide the reader and improve clarity, additional explanations about the organization and structure of the manuscript have been added to the Introduction (lines 88–94 of the new version of the paper), prior to the detailed description of the individual sections.

**Additional comments**

1. Title Clarity: As also noted by Referee 1, the current title is misleading. I recommend revisiting and refining it to better reflect the paper's focus and content, i.e remove the second part on impact on a downstream turbine and change 'multi-fidelity'.
   We thank the reviewer. We agree that the current title may not fully reflect the focus of the manuscript. For clarity and accuracy, we have revised the title to better represent the content of the paper. Now it reads: "Actuator line URANS-to-LES comparison of single and tandem floating offshore wind turbines"

2. Literature Review: The literature review could be expanded to include additional relevant studies, which you may have a look at, such as Li, Yu, Sarlak in Wind Energy (2024) Hubert, Conan, Aubrun in Wind Energy Science (2024) Messmer, Peinke, Croce, Holling in Journal of Fluid Mechanics (2025) Liu, Zhong, Zhao, Wan in Physics of Fluids (2025) Mian, Messmer, Stoevesandt, Siddiqui in Energy (2025) which may provide further context or support for your results and analysis.
   We thank the reviewer for the suggestions. The first reference mentioned by the reviewer (Li, Yu, Sarlak, Wind Energy 2024) was already included in the manuscript, while the additional studies have now been incorporated into the literature review.

3. Figure 4: The plot shows a noticeable mean offset. Could the authors provide an explanation for this discrepancy? How precisely is the reference velocity U0 determined in the experiments? While the dynamics appear well-captured, the offset remains unclear—why?
   The authors do not believe that the offset in the mean thrust and torque values is due to differences in the freestream velocity, as the offset exhibits opposite signs for thrust and torque. The load validation for both the surge and pitch cases was first performed in a previous study by the same authors (Sanvito, et al. "A novel vortex-based velocity sampling method for the actuator-line modeling of floating offshore wind turbines in windmill state." *Renewable Energy* 231 (2024): 120927.), in which the URANS simulations showed good agreement with the OC6 experimental campaign as well as with other numerical models of different fidelity levels. The new numerical simulations of the same surge and pitch cases are consistent with the results reported in that earlier work, demonstrating coherence in the numerical predictions. In addition, as reported in the present paper, the mean thrust and torque values obtained in the pitch case are in very good agreement with the ALM simulations presented in Pagamonci et al. (2025), where the same platform motion was considered. Overall, these comparisons indicate consistency and robustness in the numerical results obtained.

4. Figure 10: Please add the experimental point if you have (the figures 9 and 10 could be compiled)
   The available experimental data have been included. For clarity, the figures were kept separate but presented as subfigures within a single figure to maintain conciseness while keeping the information clear for the reader.

5. Figure 11: Please clarify what constitutes "a sufficient number of platform motion cycles" for ensuring a stable trend estimate. Additionally, linking the platform motion signal with the phase of coherent fluctuations could provide deeper insight.
A more detailed explanation has been added to the manuscript to clarify the criterion used to define a sufficient number of platform-motion cycles for phase averaging (lines 305-309 of the revised paper). Furthermore, the platform-displacement signal has been included in Figure 11 as a reference, and an additional comment has been inserted in the text (lines 313-319 of the revised paper).

6. Figure 12: Please specify the unit used in this figure.
The unit of measurement has been added.

7. Figure 12b (LES): The observation that the magnitude of pitch-related coherent fluctuations increases after 4D—despite decaying between 2D and 4D—is unexpected. What is happening there?
We thank the reviewer for this comment. As described in the previous version of the manuscript, the earlier plot was obtained by computing the amplitude of the phase-averaged streamwise velocity at $y/R=-0.75$ as the amplitude of the harmonic at the platform-motion frequency, using a Fourier transform applied to the phase-averaged velocity signal. When applied to LES and experimental data, this approach may still include small turbulent contributions at the frequency of the platform-motion (and its harmonics) , which are not completely removed by the phase-averaging procedure.
To consider a fair comparison among LES, experimental data and URANS, Fig. 12 has been updated to show the amplitude of the oscillations at the frequency of the platform motion obtained by filtering the frequency contents larger than the one associated to surge motion. The updated figure presents the peak-to-peak amplitude, instead of the semi-amplitude used in the previous version.
The conclusions drawn from the revised figure remain consistent with those of the earlier version. In the pitch case, a reduced discontinuity between the points at 4D and 5D is observed. Although a slight change in slope is still present between these two locations, the absolute magnitude of the values is very small, making this difference of limited physical relevance. Overall, a decreasing trend is identified from 3D onwards, and the underestimation of the amplitude by the URANS simulations relative to the LES results remains evident.

8. Figure 14: Please comment on the large differences between LES and URANS, especially knowing that in Mian, et al. Energy (2025), URANS was used to reconstruct flow structures, from my understanding, in a satisfying way. Here, it looks like the URANS is unable to reproduce the dynamics. Please add experimental points if available.
The study by Mian et al. (Energy, 2025) differs from the present work in several key aspects, which make a direct comparison non-trivial. In particular:
   - a different turbine representation is adopted: blade-resolved in Mian et al. versus an Actuator Line Model in the present study;

- the inflow turbulence level is much lower in Mian et al. (0.3%) compared to the present work (1.5%);
- a different turbulence model is employed: Reynolds Stress Model in Mian et al. versus a realizable $k$-$\varepsilon$ model here;
- the operating conditions and motion characteristics differ significantly.

In the present paper, a standard URANS setup was selected, relying on a turbulence model that is widely used in engineering applications, with the aim of assessing its performance in the analysed working conditions. As shown in Fig. 14, this standard URANS configuration tends to underestimate the amplitude of the motion-induced wake structures, in line with the trends already identified in the earlier analyses presented in the paper.

The experimental data at 5D are not available.

9. Figures 15 and 16: These could be merged into a single figure.

We thank the reviewer for the suggestion. However, since each of these figures already contains multiple subplots, we believe that merging them would reduce readability and make the analysis less clear for the reader. For this reason, we have chosen to keep them as two separate figures, which gives a more organized display of the results.

10. Figure 17: These results are of interest, particularly in Figure 17b. It is nice to see the difference between laminar and turbulent inflow in terms of wake meandering for surge. Focusing on the surge case (LES, turbulent inflow) around 2D, it looks like surge motion influences near-wake dynamics differently than fixed, and that the wake motions are a coupling between meandering and a mild plusating pattern. This could further be investigated and would add value to the paper. For instance, one can extend the analysis of figure 18 at different wake regions, for x ∈ [1,5]D.

Following the reviewer's suggestion, the analysis of Fig. 18 has been extended to positions between 1D and 4D to determine whether the evolution of the phenomenon differs between the fixed-bottom and surge cases at locations closer to the turbine, with a transition between the characteristic frequencies of surge motion and wake meandering identified at 5D. As detailed in lines 429–437 of the revised manuscript and illustrated in the new figure (Fig. 18), even at positions closer to the turbine, the wake dynamics still appear to be dominated by inflow turbulence, resulting in minimal differences between the fixed-bottom and surge cases.

11. Figure 18: The meandering motion observed in the wake raises questions about the mechanisms triggering it. You could examine the free-stream power spectrum to see if the region around f = 0.8 Hz contains significant energy in the inflow, potentially linking free-stream structures to the wake dynamics.

[Figure]

|  a)  PSD Ux  | b) PSD Uy | c) PSD Uz |

The figure shows the PSD spectra of the three velocity components along a horizontal line located 2D upstream of the turbine. From the spectral analysis, no dominant contribution is observed at the 0.8 Hz frequency, although some peaks are present in the region around 1 Hz. It should be noted that the colorbar thresholds used in this figure differs from that of Fig. 18 in the paper. Since the upstream spectra do not exhibit features that can be rigorously linked to the wake dynamics, the authors preferred not to include an interpretation that does not seem supported by a clear evidence.

12. Figures 19 and 20: The analysis on wake centre dynamics effectively demonstrates that surge and pitch motions do not significantly increase meandering with the turbulent inflow, nice result. These figures could be merged for conciseness.
The figures have been merged as suggested.

13. Figure 21 (URANS):The wake dynamics seem to retain some structural features of the motion, even though the inflow is turbulent, particularly the alternating blue and red regions of Uy, which suggest wake pulsating. Could the authors discuss this further, especially in relation to my previous comment 10.
The authors agree with the reviewer that the URANS wake exhibits coherent structures originated by the platform motion. This behaviour is expected and is already confirmed by the previous analyses presented in the paper, which show that, although underestimated in amplitude, the URANS simulations are able to capture motion-induced deterministic wake structures.
In Fig. 21, these structures are clearly visible in the Ux field and, consistently, they also appear in the Uy field. What emerges from the Uy distribution is that the dominant structures are primarily linked to the platform motion in spite of wake meandering, which is instead observed in the turbulent LES results.
Therefore, the behaviour shown in Fig. 21 is considered consistent with the findings previously discussed and with the interpretation provided in the manuscript.

14. Downstream turbine: The differences between LES and URANS on the loads should be linked to the results of the flow structures found in the wake, depending on the numerical method used.
In the section devoted to the analysis of the downstream turbine loads, the main differences between the LES and URANS loads are explicitly highlighted and consistently linked to the wake dynamics discussed in the previous section. In

particular, at lines 495-499 of the revised manuscript, it is noted that the phase shift observed in the thrust response is coherent with the phase shift identified in the wake oscillation analysis. Similarly, the difference in thrust amplitude between LES and URANS is shown to be consistent with the differences in wake oscillation amplitudes already observed for the single-turbine case (lines 499-500 of the revised manuscript). Furthermore, at lines 515-517 of the revised manuscript, the larger amplitude of the angle-of-attack fluctuations at the root is directly related to the spatial distribution of wake oscillation amplitudes analysed in the previous section. These links demonstrate that the discrepancies in the downstream turbine loads can be traced back to the different wake structures and dynamics reproduced by the two numerical approaches. With respect to the impact of wake meandering on the downstream turbine loads, preliminary analyses indicate that the LES simulations exhibit low-frequency contributions in the load spectra of the downstream turbine, but such oscillations occur at frequencies not exactly matching the one characteristic of the meandering observed in the previous section of the paper. However, the authors consider that a dedicated and more detailed investigation is required to determine whether these low-frequency contributions can be directly attributed to wake meandering or rather to a general interaction between the unsteady wake and the downstream turbine itself. Such an analysis represents a natural extension of the present work, aimed at clarifying how wake dynamics characterized by specific frequencies interact with a downstream turbine and how those frequency contents ultimately affect the load response. In this regard, a reference to this future work has been added in the Conclusions section.